# Malleable ribonucleoprotein machine: protein intrinsic disorder in the *Saccharomyces cerevisiae* spliceosome

Maria de Lourdes Coelho Ribeiro[1,2], Julio Espinosa[2], Sameen Islam[2], Osvaldo Martinez[2], Jayesh Jamnadas Thanki[2], Stephanie Mazariegos[2], Tam Nguyen[2], Maya Larina[3], Bin Xue[2] and Vladimir N. Uversky[2,4,5]

[1] Cancer Imaging Metabolism, H. Lee Moffitt Cancer Center & Research Institute, United States
[2] Department of Molecular Medicine, University of South Florida, Tampa, Florida, United States
[3] College of Medical Biochemistry, Volgograd State Medical University, Russia
[4] USF Health Byrd Alzheimer's Research Institute, University of South Florida, Tampa, Florida, United States
[5] Laboratory of New Methods in Biology, Institute for Biological Instrumentation, Russian Academy of Sciences, Pushchino, Moscow Region, Russia

Corresponding author
Vladimir N. Uversky,
vuversky@health.usf.edu

## ABSTRACT

Recent studies revealed that a significant fraction of any given proteome is presented by proteins that do not have unique 3D structures as a whole or in significant parts. These intrinsically disordered proteins possess dramatic structural and functional variability, being especially enriched in signaling and regulatory functions since their lack of fixed structure defines their ability to be involved in interaction with several proteins and allows them to be re-used in multiple pathways. Among recognized disorder-based protein functions are interactions with nucleic acids and multi-target binding; i.e., the functions ascribed to many spliceosomal proteins. Therefore, the spliceosome, a multimegadalton ribonucleoprotein machine catalyzing the excision of introns from eukaryotic pre-mRNAs, represents an attractive target for the focused analysis of the abundance and functionality of intrinsic disorder in its proteinaceous components. In yeast cells, spliceosome consists of five small nuclear RNAs (U1, U2, U4, U5, and U6) and a range of associated proteins. Some of these proteins constitute cores of the corresponding snRNA-protein complexes known as small nuclear ribonucleoproteins (snRNPs). Other spliceosomal proteins have various auxiliary functions. To gain better understanding of the functional roles of intrinsic disorder, we have studied the prevalence of intrinsically disordered proteins in the yeast spliceosome using a wide array of bioinformatics methods. Our study revealed that similar to the proteins associated with human spliceosomes (*Korneta & Bujnicki, 2012*), proteins found in the yeast spliceosome are enriched in intrinsic disorder.

## INTRODUCTION

Eukaryotic genes are typically characterized by a mosaic architecture, being organized into a line of alternating exons and introns. The EXONs are those EXpressed regiONs that become the mRNA, and the INTRONs are those INTRagenic regiONs that are located inside the gene and are removed in the process of making a mature messenger RNA (mRNA) from its precursor (pre-mRNA). Therefore, the process of eukaryotic mRNA maturation includes a very important step of splicing, which takes place after or concurrently with pre-mRNA transcription, and which ensures that introns are removed and exons are joined. Here, the pre-mRNA is spliced at splice junctions found at the extreme ends of each and every intron. Although some exons are constitutively spliced; i.e., they are present in every mRNA produced from a given pre-mRNA, there are multiple ways of how exons are joined during the RNA splicing, and many pre-mRNAs are alternatively spliced to generate variable forms of mRNA from a single pre-mRNA species.

Alternative (or differential) splicing is very ubiquitous in eukaryotes (e.g., ~95% of multiexonic genes in humans are alternatively spliced (*Pan et al., 2008*)), where it is believed to contribute to the greatly increased biodiversity of proteins that can be encoded by the genome (*Black, 2003*). In fact, since the different mRNAs generated from a single pre-mRNA can be translated into different protein isoforms, a single gene may code for multiple proteins. For example, >500 isoforms of the calcium-activated potassium channel Slo that are translated from the different mRNAs produced by the alternative splicing of a single *slo* gene define the ability of ears to detect a remarkable range of frequencies (*Black, 1998*; *Graveley, 2001*; *Xu et al., 2007*). The *Drosophila melanogaster* gene *Dscam* (a drosophila homolog of human Down syndrome cell adhesion molecule, DSCAM) could potentially have 38,016 splice variants which are crucial for the specificity of neuronal connectivity (*Schmucker et al., 2000*; *Celotto & Graveley, 2001*; *Kreahling & Graveley, 2005*). In human titin, which is an extremely large elastic protein (>4,200 kDa) found in heart and skeletal muscle, over a million splice pathways can be potentially derived from the PEVK region alone (so called for its high content of proline (P), glutamate (E), valine (V), and lysine (K) residues) (*Wang, 1996*; *Maruyama, 1997*; *Gregorio et al., 1999*; *LeWinter et al., 2007*; *Guo et al., 2010*). Therefore, alternative splicing defines the increased diversity of eukaryotic proteomes compared to their corresponding genomes (*Nilsen & Graveley, 2010*). Also, aberrant pre-mRNA splicing constitutes the basis of some human diseases or contributes to the severity of other human maladies (*Novoyatleva et al., 2006*; *Ward & Cooper, 2010*).

Pre-mRNA splicing takes place in all eukaryotic organisms investigated to date, from yeast to metazoans. Although in some organisms splicing might occur spontaneously, where the pre-mRNA acts as a ribozyme, being able to fold on itself, cleave itself, and then remove the intron by itself, for the majority of eukaryotic introns, splicing of pre-mRNA is done in a series of reactions catalyzed by the multimegadalton ribonucleoprotein (RNP) complex known as spliceosome (*Brow, 2002*; *Wahl, Will & Luhrmann, 2009*). The canonical assembly of the spliceosome occurs anew on each pre-mRNA that contains specific sequence elements (such as the 5' end splice, the branch point sequence, the

polypyrimidine tract, and the 3' end splice site) that are recognized and utilized during spliceosome assembly.

There are two spliceosome types, the major spliceosome, which contains five small nuclear ribonucleoproteins (snRNPs, often pronounced as snurps, the U1, U2, U4/U6, and U5 snRNPs) as the main building blocks, and which is responsible for removing the vast majority of pre-mRNA introns; and the minor spliceosome, which is present in some metazoan species and plants, and which is composed of the compositionally distinct but functionally analogous U11/U12 and U4atac/U6atac snRNPs, with the U5 snRNP shared between the machineries (*Patel & Steitz, 2003*). The major spliceosome is composed of five small nuclear RNA (snRNA) molecules: U1, U2, U4, U5 and U6, and a number of core proteins. A common feature of all spliceosomal snRNPs except U6 is the presence of seven mutually related Sm proteins. U6 contains a set of related "like-Sm" (Lsm) proteins (*Veretnik et al., 2009*). In the spliceosomal snRNPs, the Sm or Lsm proteins form a ring structure whereas a U-rich sequence in the snRNA binds in the positively charged central hole of this ring (*Kambach, Walke & Nagai, 1999*; *Kambach et al., 1999*). This core structure is further enhanced by 80–150 proteins that are abundant in the human spliceosome and are essential to the process of spliceosome-dependent splicing (*Agafonov et al., 2011*).

Based on the proteomic analysis of yeast spliceosome it has been concluded that the yeast splicing machinery likely contains the evolutionarily conserved core set of spliceosomal proteins that are required for constitutive splicing (*Fabrizio et al., 2009*). On the other hand, the number of proteins found in the yeast B, B$^{act}$and C complexes was noticeably lower than that in the corresponding metazoan complex (*Fabrizio et al., 2009*; *Will & Luhrmann, 2011*). For example, there were only ∼60 proteins in yeast pre-catalytic B complexes (compared to ∼110 in humans and *D. melanogaster* spliceosomes), including essentially all U1, U2, and U4/U6.U5 tri-snRNP proteins together with proteins of the nineteen complex (NTC) and mRNA retention and splicing (RES) complex (*Fabrizio et al., 2009*). Similarly, yeast C complexes contained only ∼50 proteins compared to ∼110 in metazoan C complexes. Therefore, this analysis revealed that yeast spliceosomes contain ∼90 proteins, almost all of which have homologs in higher eukaryotes (*Fabrizio et al., 2009*). Many of the remaining ∼80 proteins found in human and *D. melanogaster* spliceosomes but not detected in yeast were shown to play a role in alternative splicing, a process that is essentially absent in yeast (*Fabrizio et al., 2009*). The much lower number of proteins in yeast spliceosome compared to the metazoan counterpart suggests that yeast possesses a different, or at least simplified splicing mechanism. For example, it is likely that this reduction can be related to the extremely low number of spliceable genetic material (there are only about 250 introns in *S. cerevisiae*).

The highly dynamic conformation and composition of the spliceosomal proteins determine the accuracy and flexibility of the splicing machinery (*Will & Luhrmann, 2011*). The major constituents and regulators of the spliceosome (snRNPs and related non-snRNP proteins) are mostly conserved from yeast to metazoan (*Fabrizio et al., 2009*). In yeast, the spliceosome assembly on its pre-mRNA substrate represents a highly ordered and regulated process that starts with recognition of the 5' end of the intron (5' splice site,

5'ss) of the pre-mRNA by the U1 snRNP. Next, the U2 snRNP binds to the pre-mRNA's branch site, forming complex A. This complex A then binds the preformed U4/U6.U5 tri-snRNP to produce penta-snRNP complex B, which contains a full set of five snRNAs in a pre-catalytic state. Complex B is then activated for catalysis by a major rearrangement of its RNA network and by global changes of its overall structure, where the association of U4 with U6 is destabilized, enabling U6 to isomerize into a base-pairing interaction with U2 to form part of the catalytic center of the spliceosome. This remodeling also includes dissociation of the U1 and U4 snRNAs and binding of a set of specific proteins leading to the formation of the activated spliceosome ($B^{act}$). Step 1 of splicing takes place in catalytically activated complex, $B^*$. Here, the adenosine at the branch site attacks the 5'ss site of the pre-mRNA, generating a cleaved 5'-exon and intron-3'-exon intermediate. Finally, the complex C is formed via binding another set of specific proteins. This complex C catalyzes step 2 of splicing, in which the intron is cleaved at the 3'-splice-site (3'ss) with concomitant ligation of the 5' and 3' exons (*Fabrizio et al., 2009*; *Will & Luhrmann, 2011*).

Importantly, although the RNA acts as a catalyst in snRNPs, the spliceosomal proteins are not just passive building blocks that hold the RNA in the correct configuration to stabilize it, but carry out essential recognition and catalytic functions during the assembly of the spliceosome and splicing-related catalytic reactions (*Abelson, 2008*; *Pyle, 2008*; *Fabrizio et al., 2009*), and also play a crucial role in the selection of intron substrates during the alternative splicing (*Caceres & Kornblihtt, 2002*). It is also important to remember that in addition to the five snRNAs, pre-mRNA splicing requires the activity of a large number of proteins, often called pre-mRNA processing proteins (Prps). Many spliceosomal and non-spliceosomal proteins are believed to have important activities related to the specificity, accuracy, and regulation of the spliceosome (*Russell et al., 2000*). Since these proteins are involved in numerous protein–protein and protein-RNA interactions, there is a great chance that at least some of them might belong to the class of intrinsically disordered proteins.

Intrinsically disordered proteins (IDPs) or intrinsically disordered protein regions (IDPRs) lack stable tertiary and/or secondary structure under physiological conditions *in vitro* (*Wright & Dyson, 1999*; *Uversky, Gillespie & Fink, 2000*; *Dunker et al., 2001*; *Dunker & Obradovic, 2001*; *Dunker et al., 2002*; *Dunker, Brown & Obradovic, 2002*; *Dyson & Wright, 2002*; *Tompa, 2002*; *Uversky, 2002a*; *Uversky, 2002b*; *Uversky, 2003*; *Tompa & Csermely, 2004*; *Daughdrill et al., 2005*; *Dunker et al., 2005*; *Dyson & Wright, 2005*; *Oldfield et al., 2005a*; *Tompa, 2005*; *Tompa, Szasz & Buday, 2005*; *Uversky, Oldfield & Dunker , 2005*; *Radivojac et al., 2007*; *Vucetic et al., 2007*; *Xie et al., 2007a*; *Xie et al., 2007b*; *Cortese, Uversky & Dunker, 2008*; *Dunker et al., 2008a*; *Dunker et al., 2008b*; *Dunker & Uversky, 2008*; *Oldfield et al., 2008*; *Russell & Gibson, 2008*; *Tompa & Fuxreiter, 2008*; *Uversky, Oldfield & Dunker, 2008*; *Tompa et al., 2009*; *Wright & Dyson, 2009*; *Uversky & Dunker, 2010*). They are highly abundant in nature, with ∼25%–30% of eukaryotic proteins being mostly disordered, and with >50% of eukaryotic proteins and >70% of signaling proteins having long disordered regions (*Dunker et al., 2000*; *Ward et al., 2004*; *Uversky, 2010*; *Schad, Tompa & Hegyi, 2011*; *Xue, Dunker & Uversky, 2012*). Functional repertoire of

IDPs is very broad and complements functions of ordered proteins, and functions of IDPs may arise from the specific disorder form, from inter-conversion of disordered forms, or from transitions between disordered and ordered conformations (*Dunker et al., 2001*; *Dunker & Obradovic, 2001*; *Uversky, 2002a*; *Uversky, 2002b*; *Uversky & Dunker, 2010*). The choice between these conformations is determined by the peculiarities of the protein environment, and many IDPs possess an exceptional ability to fold in a template dependent manner, where a single IDPR can bind to multiple partners gaining very different structures in the bound state (*Oldfield et al., 2008*; *Hsu et al., 2012*). Often, IDPs are involved in regulation, signaling and control pathways, where binding to multiple partners and high-specificity/low-affinity interactions play a crucial role and where IDPs/IDPRs play different roles in regulation of the function of their binding partners and in promotion of the assembly of supra-molecular complexes (*Wright & Dyson, 1999*; *Dunker et al., 2001*; *Dunker et al., 2002*; *Dunker, Brown & Obradovic, 2002*; *Dyson & Wright, 2002*; *Dunker et al., 2005*; *Dyson & Wright, 2005*; *Uversky, Oldfield & Dunker, 2005*; *Cortese, Uversky & Dunker, 2008*; *Dunker et al., 2008a*; *Dunker et al., 2008b*; *Dunker & Uversky, 2008*; *Oldfield et al., 2008*; *Uversky & Dunker, 2010*). In a bioinformatics analysis performed in 2008, it was found that out of the 711 Swiss-Prot functional keywords associated with at least 20 proteins, 262 were strongly positively correlated with long intrinsically disordered regions, and 302 were strongly negatively correlated (*Vucetic et al., 2007*; *Xie et al., 2007a*; *Xie et al., 2007b*).

IDPs and IDPRs are the key players in various protein–protein interaction networks, being especially abundant among hub proteins and their binding partners (*Dunker et al., 2005*; *Dosztanyi et al., 2006*; *Ekman et al., 2006*; *Haynes et al., 2006*; *Patil & Nakamura, 2006*; *Singh et al., 2006*). Furthermore, regions of pre-mRNA which undergo alternative splicing commonly encode for the disordered regions (*Romero et al., 2006*). This association of alternative splicing and intrinsic disorder helps proteins to avoid folding difficulties and provides a novel mechanism for developing tissue-specific protein interaction networks (*Romero et al., 2006*; *Uversky, Oldfield & Dunker, 2008*).

The hypothesis that the spliceosomal proteins might be enriched in intrinsic disorder is supported by the aforementioned results of the bioinformatics analysis of the correlation between the Swiss-Prot functional keywords and protein intrinsic disorder which clearly showed that mRNA processing and mRNA splicing were among 20 top biological processes associated with protein intrinsic disorder (*Xie et al., 2007a*). Furthermore, the functional keyword spliceosome was at the position #4 of the top 20 cellular components strongly correlated with predicted disorder (*Vucetic et al., 2007*). Also, there are several case studies, where intrinsic disorder was found in some spliceosomal proteins. For example, NMR analysis revealed that the flanking N- (residues 1–20) and C-terminal regions (residues 100–125) of the protein p14 (which is a subunit of the essential splicing factor 3b (SF3b) present in both the major and minor spliceosomes (*Will et al., 1999*; *Will et al., 2001*; *Will et al., 2004*), and which is located near the catalytic center of the spliceosome and is responsible for the first catalytic step of the splicing reaction (*Query, Strobel & Sharp, 1996*; *Will et al., 2004*)) are unstructured (*Spadaccini et al., 2006*). Serine/arginine-rich

(SR) splicing factors are important spliceosomal IDPs, which, besides their significance for both constitutive and alternative splicing (*Zahler et al., 1992*), play key roles in the spliceosome assembly by facilitating recruitment of components of the spliceosome via protein–protein interactions (*Roscigno & Garcia-Blanco, 1995*) that are potentially mediated by the disordered SR domains of these splicing factors (*Haynes & Iakoucheva, 2006*). Finally, a recently reported systematic bioinformatics analysis of the abundance of intrinsic disorder in the proteome of the human spliceosome provided a strong support to the "disordered spliceosome" hypothesis (*Korneta & Bujnicki, 2012*).

Since metazoan spliceosomes are rather different from the yeast counterparts (for example, yeast spliceosomes have radically fewer proteins than metazoan spliceosomes, possessing typically less than half proteins per spliceosomal complex (*Fabrizio et al., 2009*)), and since the protein sequence homology between yeast and human spliceosomal proteins ranges from 36 to a little over 50% (*Ben-Yehuda et al., 2000*), data on the abundance of intrinsic disorder in human spliceosomal proteome cannot be directly projected to the yeast proteomes. Therefore, in the present work we have studied the prevalence of intrinsic disorder in the yeast spliceosome using a wide array of bioinformatics methods. Our study showed that similar to the proteins associated with human spliceosomes (*Korneta & Bujnicki, 2012*), proteins found in the yeast spliceosome are relatively enriched in intrinsic disorder.

## MATERIALS AND METHODS

### Dataset

In this work we studied the presence of intrinsic disordered proteins (IDP) in the yeast spliceosome. The first step was to search of the UniProt database (http://www.uniprot.org) for known proteins in the baking yeast's (*Saccharomyces cerevisiae*) spliceosome. This query resulted in 140 proteins, from which 109 reviewed entries were selected to make sure that the proteins chosen for analysis were manually annotated and reviewed by UniProtKB curators. The amino acid sequences in FASTA format of all these 109 yeast spliceosomal proteins were retrieved from the UniProt database and used in subsequent analysis.

At the next stage, we compared this dataset with a set of yeast spliceosomal proteins found via the comprehensive proteomic analysis of the yeast spliceosomal complex B, activated B[act], and step 1 complex C (*Fabrizio et al., 2009*). This experimentally determined set contained 89 proteins directly assigned to different spliceosomal components and complexes. Table 1 groups these proteins according to their functional/structural annotations and also lists 20 extra spliceosomal proteins found via the UniProt search.

### Analysis of the amino acid composition of yeast spliceosomal proteins

To gain insight into the relationships between sequence and disorder, amino acid compositions of different datasets were compared using an approach recently developed for IDPs (*Dunker et al., 2001*; *Vacic et al., 2007a*). To this end, the fractional difference

**Table 1** Major structural characteristics and disorder propensities of the proteins from the *Saccharomyces cerevisiae* spliceosome analyzed in this study.

| Yeast protein name | Gene name | UniProt ID | PDB ID (residues) | MW (kDa) | Length | PONDR-FIT score | PONDR VLXT score | FoldIndex score | RONN score | MoRFs | AIBSs | Human protein name |
|---|---|---|---|---|---|---|---|---|---|---|---|---|
| **Sm proteins** | | | | | | | | | | | | |
| B | YER029C | P40018 | | 22.4 | 196 | 0.64 | 0.65 | 0.73 | 0.63 | 167–184 | 79–94, 102–109, 134–196 | B |
| D1 | YGR074W | Q02260 | | 16.3 | 146 | 0.36 | 0.54 | 0.29 | 0.47 | 92–109 | | D1 |
| D2 | YLR275W | Q06217 | | 12.9 | 110 | 0.32 | 0.40 | 0.14 | 0.18 | | | D2 |
| D3 | YLR147C | P43321 | | 11.2 | 101 | 0.33 | 0.48 | 0.28 | 0.17 | | 66–79 | D3 |
| E | YOR159C | Q12330 | | 10.4 | 94 | 0.23 | 0.31 | 0.00 | 0.15 | | | E |
| F | YPR182W | P54999 | 1N9R (1–86) | 9.66 | 86 | 0.22 | 0.26 | 0.00 | 0.19 | | | F |
| G | YFL017W-A | P40204 | | 8.46 | 77 | 0.34 | 0.39 | 0.00 | 0.00 | | | G |
| **U1 snRNP proteins** | | | | | | | | | | | | |
| Prp39 | YML046W | P39682 | | 77.7 | 629 | 0.04 | 0.05 | 0.21 | 0.07 | | | |
| Snu71 | YGR013W | P53207 | | 71.4 | 620 | 0.45 | 0.36 | 0.68 | 0.48 | 270–287, 316–333, 379–396, 407–424 | 3–8, 275–295, 315–327, 372–384, 387–402, 405–428, 450–474, 495–507, 566–577 | S164 |
| Prp40 | YKL012W | P33203 | 1O6W (1–75) 2B7E (134–189) 2KFD (489–552) | 69.1 | 583 | 0.13 | 0.26 | 0.86 | 0.35 | | 93–99, 497–506 | FBP11 |
| Prp42 | YDR235W | Q03776 | | 65.1 | 544 | 0.05 | 0.07 | 0.21 | 0.04 | 526–543 | | |
| Nam8 | YHR086W | Q00539 | | 57.0 | 523 | 0.44 | 0.37 | 0.19 | 0.46 | 1–18 | 1–13, 39–44, 227–234, 440–446 | T1A1/TIAR |
| Snu56 | YDR240C | Q03782 | | 56.5 | 492 | 0.25 | 0.22 | 0.47 | 0.24 | 9–26, 345–362 | 1–16, 42–49, 320–331, 342–347 | |

| Yeast protein name | Gene name | UniProt ID PDB ID (residues) | MW (kDa) | Length | PONDR-FIT score | PONDR VLXT score | FoldIndex score | RONN score | MoRFs | AIBSs | Human protein name |
|---|---|---|---|---|---|---|---|---|---|---|---|
| Snp1 | YIL061C | Q00916 | 34.4 | 300 | 0.44 | 0.50 | 0.77 | 0.49 | 1-18<br>53-70<br>211-228<br>240-257 | 1-12<br>44-53<br>245-283 | U1-70K |
| Mud1 | YBR119W | P32605 | 34.3 | 298 | 0.46 | 0.56 | 0.57 | 0.27 | 119-136<br>222-239 | 1-6<br>155-167 | U1-A |
| Luc7 | YDL087C | Q07508 | 30.2 | 261 | 0.17 | 0.38 | 0.51 | 0.43 | 14-31 | 45-51<br>201-207<br>222-229 | LUC7B1 |
| Yhc1 | YLR298C | Q05900 | 27.1 | 231 | 0.64 | 0.43 | 0.89 | 0.48 | 71-88<br>185-202 | 25-46<br>73-92<br>102-133<br>185-203<br>223-231 | U1-C |
| **U2 snRNP proteins** | | | | | | | | | | | |
| Rse1 | YML049C | Q04693 | 153.8 | 1,361 | 0.12 | 0.19 | 0.16 | 0.16 | 335-352 | 814-822<br>851-862 | SF3b130 |
| Hsh155 | YMR288W | P49955 | 110.0 | 971 | 0.14 | 0.23 | 0.17 | 0.16 | 3-20<br>63-80 | 1-41<br>63-74<br>124-145 | SF3b155 |
| Prp9 | YDL030W | P19736<br>4DGW (1-389) | 63.0 | 530 | 0.29 | 0.28 | 0.75 | 0.36 | 77-94<br>335-352<br>512-529 | 77-84<br>409-414<br>449-455<br>520-530 | SF3a60 |
| Cus1 | YMR240C | Q02554 | 50.3 | 436 | 0.50 | 0.52 | 0.79 | 0.60 | 14-31<br>80-97<br>192-209 | 17-36<br>53-72<br>88-98<br>143-166<br>362-372 | SF3b145 |

| Yeast protein name | Gene name | UniProt ID PDB ID (residues) | MW (kDa) | Length | PONDR-FIT score | PONDR VLXT score | FoldIndex score | RONN score | MoRFs | AIBSs | Human protein name |
|---|---|---|---|---|---|---|---|---|---|---|---|
| | | | | | | | | | 419-436 | 401-413 429-436 | |
| Prp21 | YJL203W | P32524 4DGW (87-237) | 33.1 | 280 | 0.26 | 0.29 | 0.81 | 0.27 | | 79-84 | SF3a120 |
| Prp11 | YDL043C | Q07350 4DGW (149-266) | 29.9 | 266 | 0.24 | 0.38 | 0.74 | 0.47 | | 58-76 93-100 135-141 | SF3a66 |
| Lea1 | YPL213W | Q08963 | 27.2 | 238 | 0.19 | 0.39 | 0.17 | 0.40 | | 130-140 198-204 232-238 | U2-A' |
| Hsh49 | YOR319W | Q99181 | 24.5 | 213 | 0.12 | 0.07 | 0.13 | 0.03 | | | SF3b49 |
| Msl1 | YIR009W | P40567 | 12.8 | 111 | 0.29 | 0.27 | 0.27 | 0.25 | | | U2-B'' |
| Rds3 | YPR094W | Q06835 2K0A (2-107) | 12.3 | 107 | 0.17 | 0.38 | 0.00 | 0.04 | | | SF3b14b |
| Ysf3 | YNL138W-A | P0C074 | 10.0 | 85 | 0.41 | 0.29 | 0.99 | 0.32 | | 12-19 | SF3b10 |
| U5 snRNP proteins | | | | | | | | | | | |
| Prp8 | YHR165C | P33334 3SBG (1836-2397) 3E66 (1822-2095) | 279.5 | 2,413 | 0.13 | 0.26 | 0.40 | 0.21 | 235-252 | 1-20 28-49 58-72 83-109 160-174 | 220K |
| Brr2 | YER172C | P32639 3IM1 (1839-2163) | 246.2 | 2,163 | 0.10 | 0.17 | 0.23 | 0.18 | 24-41 | 11-34 56-71 102-110 139-148 174-192 205-212 227-232 285-291 | 200K |
| Snu114 | YKL173W | P36048 | 114.0 | 1,008 | 0.14 | 0.24 | 0.18 | 0.17 | 1-18 61-78 | 1-21 53-77 485-491 543-550 | 116K |

Table 1 (*continued*)

| Yeast protein name | Gene name | UniProt ID | PDB ID (residues) | MW (kDa) | Length | PONDR-FIT score | PONDR VLXT score | FoldIndex score | RONN score | MoRFs | AIBSs | Human protein name |
|---|---|---|---|---|---|---|---|---|---|---|---|---|
| Prp6 | YBR055C | P19735 | | 104.2 | 899 | 0.18 | 0.33 | 0.29 | 0.22 | 15-32 63-80 105-122 144-161 172-189 200-217 | 1-34 45-55 68-81 100-121 145-158 199-205 | 102K |
| Prp28 | YDR243C | P23394 | | 66.6 | 588 | 0.16 | 0.21 | 0.31 | 0.24 | 6-23 60-77 | 1-17 27-35 62-69 109-120 | 100K |
| Lin1 | YHR156C | P38852 | | 40.4 | 340 | 0.45 | 0.42 | 0.66 | 0.52 | 1-17 31-48 63-80 94-111 | 1-11 25-48 89-111 122-140 | 52K |
| Dib1 | YPR082C | Q06819 | | 16.8 | 143 | 0.11 | 0.12 | 0.06 | 0.00 | | | 15K |
| U4/U6 snRNP proteins | | | | | | | | | | | | |
| Prp31 | YGR091W | P49704 | | 56.3 | 494 | 0.36 | 0.36 | 0.39 | 0.41 | 17-34 338-355 383-400 438-455 470-487 | 301-312 371-388 404-423 434-463 468-477 487-494 | 61K |
| Prp3 | YDR473C | Q03338 | | 55.9 | 469 | 0.48 | 0.45 | 0.87 | 0.47 | 15-32 51-68 109-126 157-174 218-235 272-289 336-353 | 16-27 39-65 85-98 228-237 258-265 274-281 336-344 | 90K |

Table 1 (*continued*)

| Yeast protein name | Gene name | UniProt ID PDB (residues) | MW (kDa) | Length | PONDR-FIT score | PONDR VLXT score | FoldIndex score | RONN score | MoRFs | AIBSs | Human protein name |
|---|---|---|---|---|---|---|---|---|---|---|---|
| Prp4 | YPR178W | P20053 | 52.4 | 465 | 0.22 | 0.31 | 0.32 | 0.22 | 71-88, 110-127, 149-166 | 1-8 | 60K |
| Snu13 | YEL026W | P39990 2ALE (1-126) | 13.6 | 126 | 0.14 | 0.18 | 0.00 | 0.05 | | | 15.5K |

U4/U6.U5 snRNP proteins

| Yeast protein name | Gene name | UniProt ID PDB (residues) | MW (kDa) | Length | PONDR-FIT score | PONDR VLXT score | FoldIndex score | RONN score | MoRFs | AIBSs | Human protein name |
|---|---|---|---|---|---|---|---|---|---|---|---|
| Snu66 | YOR308C | Q12420 3PLU (6-24) 3PLV (37-57) | 66.4 | 587 | 0.76 | 0.70 | 0.94 | 0.81 | 19-37, 87-104, 146-163, 228-245, 251-269, 299-316, 367-384, 400-417, 446-463, 504-521, 538-555 | 17-28, 44-58, 77-99, 109-119, 141-156, 160-169, 208-215, 228-247, 256-264, 275-282, 299-310, 360-386, 439-466, 496-506, 523-534, 542-554, 564-569, 580-587 | 110K |
| Sad1 | YFR005C | P43589 | 52.2 | 448 | 0.08 | 0.10 | 0.22 | 0.26 | | | 65K |
| Spp381 | YBR152W | P38282 | 33.8 | 291 | 0.87 | 0.69 | 1.00 | 0.82 | 1-18, 36-53, 113-130, 151-168, 215-232, 264-281 | 1-17, 26-54, 93-158, 166-175, 230-238 | |
| Prp38 | YGR075C | Q00723 | 28.0 | 242 | 0.21 | 0.15 | 0.25 | 0.26 | | | hPRP38 |

(*continued on next page*)

| Yeast protein name | Gene name | UniProt ID PDB ID (residues) | MW (kDa) | Length | PONDR-FIT score | PONDR VLXT score | FoldIndex score | RONN score | MoRFs | AIBSs | Human protein name |
|---|---|---|---|---|---|---|---|---|---|---|---|
| Snu23 | YDL098C | Q12368 | 22.7 | 194 | 0.28 | 0.26 | 0.90 | 0.40 | 126-143 | | hSNU23/ZMAT2 |
| **Lsm proteins** | | | | | | | | | | | |
| LSm4 | YER112W | P40070 | 21.3 | 187 | 0.74 | 0.59 | 0.70 | 0.65 | 163-180 | 65-91, 106-116, 120-125, 134-187 | LSm4 |
| LSm7 | YNL147W | P53905 | 13.0 | 115 | 0.62 | 0.61 | 0.32 | 0.21 | | 26-34, 46-53 | LSm7 |
| LSm8 | YJR022W | P47093 | 12.4 | 109 | 0.18 | 0.23 | 0.28 | 0.16 | | | LSm8 |
| LSm2 | YBL026W | P38203 | 11.2 | 95 | 0.23 | 0.10 | 0.29 | 0.11 | | | LSm2 |
| LSm5 | YER146W | P40089 | 10.4 | 93 | 0.33 | 0.45 | 0.14 | 0.00 | | | LSm5 |
| LSm3 | YLR438C | P57743 3BW1 (1-89) | 10.0 | 89 | 0.32 | 0.52 | 0.38 | 0.09 | | | LSm3 |
| LSm6 | YDR378C | Q06406 | 9.38 | 86 | 0.21 | 0.20 | 0.00 | 0.27 | | | LSm6 |
| **RES proteins** | | | | | | | | | | | |
| Bud13/Cwc26 | YGL174W | P46947 | 30.5 | 266 | 0.63 | 0.44 | 0.98 | 0.73 | 1-18, 59-76 | 1-10, 36-45, 56-64, 66-75, 79-86, 104-112, 157-174, 209-214 | MGC13125 |
| Pml1 | YLR016C | Q07930 3ElV (1-204) | 23.7 | 204 | 0.16 | 0.42 | 0.56 | 0.30 | 11-28 | | SNIP1 ? |
| Ist/Snu17 | YIR005W | P40565 | 17.1 | 148 | 0.18 | 0.20 | 0.77 | 0.03 | | | CGI-79 ? |
| **NTC/Prp19 Complex** | | | | | | | | | | | |
| Syf1 | YDR416W | Q04048 | 100.2 | 859 | 0.09 | 0.18 | 0.27 | 0.10 | | | hSYF1/XAB2 |
| Clf1 | YLR117C | Q12309 | 82.4 | 687 | 0.04 | 0.18 | 0.51 | 0.07 | 32-49 | | CRNKL1 |
| Cef1 | YMR213W | Q03654 | 67.7 | 590 | 0.44 | 0.57 | 0.76 | 0.58 | 210-228, 284-301 | 94-103, 116-124, 152-160, 193-201, 212-226, 240-248, 292-303 | CDC5L |

| Yeast protein name | Gene name | UniProt ID PDB ID (residues) | MW (kDa) | Length | PONDR-FIT score | PONDR VLXT score | FoldIndex score | RONN score | MoRFs | AIBSs | Human protein name |
|---|---|---|---|---|---|---|---|---|---|---|---|
| | | | | | | | | | | 310-315 | |
| | | | | | | | | | 316-333 | 319-337 | |
| | | | | | | | | | 362-379 | 359-374 | |
| | | | | | | | | | | 427-439 | |
| | | | | | | | | | | 453-463 | |
| | | | | | | | | | 489-506 | | |
| | | | | | | | | | | 543-549 | |
| | | | | | | | | | | 558-566 | |
| Prp19 | YLL036C | P32523 1N87 (1-56) 3LRV (165-503) | 56.6 | 503 | 0.13 | 0.23 | 0.23 | 0.24 | | 1-6 | hPRP19 |
| Isy1/Ntc30 | YJR050W | P21374 | 28.0 | 235 | 0.39 | 0.48 | 1.00 | 0.52 | | 9-14 | KIAA1160 |
| | | | | | | | | | | 167-173 | |
| | | | | | | | | | | 185-190 | |
| Syf2 | YGR129W | P53277 | 24.8 | 215 | 0.84 | 0.67 | 1.00 | 0.75 | | 1-7 | GCIP p29 |
| | | | | | | | | | 63-80 | 66-78 | |
| | | | | | | | | | 96-113 | 93-113 | |
| | | | | | | | | | 151-168 | 153-169 | |
| | | | | | | | | | 187-204 | 186-198 | |
| Snt309 | YPR101W | Q06091 | 20.7 | 175 | 0.24 | 0.44 | 0.61 | 0.21 | | 58-65 | SPF27 |
| Ntc20 | YBR188C | P38302 | 16.0 | 140 | 0.34 | 0.47 | 0.94 | 0.60 | | 1-7 | |
| | | | | | | | | | 6-23 | 35-49 | |
| | | | | | | | | | | 91-112 | |
| | | | | | | | | | | 121-127 | |
| **NTC-Related Proteins** | | | | | | | | | | | |
| Prp46 | YPL151C | Q12417 | 50.7 | 451 | 0.06 | 0.17 | 0.15 | 0.26 | | 25-30 | PRL1 |
| Prp45 | YAL032C | P28004 | 42.5 | 379 | 0.32 | 0.62 | 0.82 | 0.59 | 1-18 | 32-41 | SKIP1 |
| | | | | | | | | | | 71-81 | |
| | | | | | | | | | | 103-112 | |
| | | | | | | | | | | 121-133 | |
| | | | | | | | | | | 165-172 | |
| | | | | | | | | | | 178-185 | |
| | | | | | | | | | 191-208 | 200-208 | |
| | | | | | | | | | | 225-231 | |

(continued on next page)

| Yeast protein name | Gene name | UniProt ID PDB (residues) | MW (kDa) | Length | PONDR-FIT score | PONDR VLXT score | FoldIndex score | RONN score | MoRFs | AIBSs | Human protein name |
|---|---|---|---|---|---|---|---|---|---|---|---|
| | | | | | | | | | 261-278 | 271-278, 304-310, 325-344 | |
| Ecm2 | YBR065C | P38241 | 40.9 | 364 | 0.16 | 0.26 | 0.15 | 0.21 | 1-18, 347-364 | 359-364 | RBM22 ? |
| Cwc2 | YDL209C | Q12046 3U1L (1-240) | 38.4 | 339 | 0.28 | 0.38 | 0.50 | 0.40 | 218-235, 246-263, 291-308, 319-336 | 181-187, 243-255, 284-292, 302-307, 326-334 | AD-002/HSPC148 |
| Cwc15 | YDR163W | Q03772 | 19.9 | 175 | 0.80 | 0.59 | 1.00 | 0.83 | 27-44, 96-113, 148-165 | 11-53, 93-102, 128-138, 150-175 | RBM22 |
| Bud31 | YCR063W | P25337 | 18.4 | 157 | 0.34 | 0.34 | 0.98 | 0.45 | 14-31 | | G10 |
| Early Splicing Factors | | | | | | | | | | | |
| Prp5 | YBR237W | P21372 | 96.4 | 849 | 0.43 | 0.51 | 0.48 | 0.47 | 1-18, 30-47, 103-120, 145-162, 661-678, 683-700 | 22-33, 100-107, 116-122, 156-172, 735-740 | hPRP5 |
| Urn1 | YPR152C | Q06525 2JUC (212-266) | 54.1 | 465 | 0.39 | 0.45 | 0.80 | 0.37 | 6-23 | 56-71, 104-112, 126-135, 163-174, 194-206, 216-231, 293-303 | TCERG1 |
| Known Splicing Factors | | | | | | | | | | | |
| Prp2 | YNR011C | P20095 | 99.8 | 876 | 0.18 | 0.25 | 0.30 | 0.23 | 1-6 | 1-6 | DDX16 |

| Yeast protein name | Gene name | UniProt ID | PDB ID (residues) | MW (kDa) | Length | PONDR-FIT score | PONDR VLXT score | FoldIndex score | RONN score | MoRFs | AIBSs | Human protein name |
|---|---|---|---|---|---|---|---|---|---|---|---|---|
| Cwc22 | YGR278W | P53333 | | 67.3 | 577 | 0.19 | 0.28 | 0.37 | 0.24 | 169-186<br>525-542 | 13-32<br>46-64<br>100-105<br>117-129<br>168-181<br>462-471<br>520-532<br>540-548<br>571-577 | KIAA1604 |
| Cwc27 | YPL064C | Q02770 | | 35.0 | 301 | 0.38 | 0.51 | 0.65 | 0.42 | 284-301 | 217-225<br>233-241<br>283-301 | NY-CO-10 |
| Cwc23 | YGL128C | P52868 | | 33.2 | 283 | 0.09 | 0.21 | 0.37 | 0.19 | | | DNAJ A1 ? |
| Yju2/ Cwc16 | YKL095W | P28320 | | 32.3 | 278 | 0.69 | 0.63 | 0.90 | 0.64 | 7-24<br>147-164<br>214-231<br>236-253<br>261-278 | 76-88<br>162-173<br>186-206<br>212-228<br>245-278 | CCDC130 |
| Cwc24 | YLR323C | P53769 | | 29.7 | 259 | 0.56 | 0.34 | 0.85 | 0.55 | 1-18<br>20-37<br>61-78<br>108-125 | 1-8<br>23-31<br>41-51<br>71-77 | RNF113A |
| Spp2 | YOR148C | Q02521 | | 20.6 | 185 | 0.59 | 0.40 | 0.76 | 0.77 | 1-18 | 1-8<br>104-114<br>141-171 | GPKOW/T54 |
| Cwc25 | YNL245C | P53854 | | 20.4 | 179 | 0.91 | 0.72 | 0.00 | 0.83 | 10-27<br>72-89 | 1-13<br>69-83<br>92-104<br>116-127<br>129-149<br>162-179 | CCDC49 |

| Yeast protein name | Gene name | UniProt ID | PDB ID (residues) | MW (kDa) | Length | PONDR-FIT score | PONDR VLXT score | FoldIndex score | RONN score | MoRFs | AIBSs | Human protein name |
|---|---|---|---|---|---|---|---|---|---|---|---|---|
| Cwc21 | YDR482C | Q03375 | | 15.8 | 135 | 1.00 | 0.94 | 1.00 | 1.00 | 1-18<br>99-116 | 1-27<br>44-66<br>96-106<br>117-123 | Srm300 |
| Step 2 proteins | | | | | | | | | | | | |
| Prp22 | YER013W | P24384 | | 130.0 | 1.145 | 0.18 | 0.27 | 0.34 | 0.38 | 368-385<br>388-405<br>1115-1132 | 190-200<br>230-250<br>297-316<br>319-330<br>370-407<br>446-455 | hPRP22 |
| Prp16 | YKR086W | P15938 | | 121.6 | 1.071 | 0.19 | 0.28 | 0.29 | 0.32 | 2-19<br>166-183 | 55-52<br>59-68<br>89-99<br>119-137<br>178-185<br>197-203<br>284-290<br>353-360<br>1064-1071 | hPRP16 |
| Prp17 | YDR364C | P40968 | | 52.1 | 455 | 0.23 | 0.23 | 0.57 | 0.28 | 17-34<br>118-135 | 1-9<br>91-99<br>146-151 | hPRP17 |
| Slu7 | YDR088C | Q02775 | | 44.6 | 382 | 0.57 | 0.50 | 0.99 | 0.69 | 35-52<br>111-128<br>147-164<br>170-187<br>213-230 | 14-46<br>69-87<br>95-111<br>115-126<br>177-182<br>218-226<br>298-305 | hSLU7 |

Table 1 (*continued*)

| Yeast protein name | Gene name | UniProt ID PDB ID (residues) | MW (kDa) | Length | PONDR-FIT score | PONDR VLXT score | FoldIndex score | RONN score | MoRFs | AIBSs | Human protein name |
|---|---|---|---|---|---|---|---|---|---|---|---|
| | | | | | | | | | | 327-338 | |
| | | | | | | | | | | 353-359 | |
| | | | | | | | | | | 377-382 | |
| Prp18 | YGR006W | P33411 1DVK (80–251) | 28.4 | 251 | 0.43 | 0.38 | 0.34 | 0.46 | 12-29 | 1-14 | hPRP18 |
| | | | | | | | | | | 23-37 | |
| **Disassembly proteins** | | | | | | | | | | | |
| Prp43 | YGL120C | P53131 2XAU (1–767) | 87.6 | 767 | 0.15 | 0.39 | 0.34 | 0.23 | | 1-9 | hPRP43 |
| | | | | | | | | | | 17-32 | |
| | | | | | | | | | | 42-54 | |
| | | | | | | | | | | 81-89 | |
| | | | | | | | | | | 106-112 | |
| | | | | | | | | | | 339-345 | |
| | | | | | | | | | | 385-391 | |
| Spp382 | YLR424W | Q06411 | 83.1 | 708 | 0.13 | 0.21 | 0.31 | 0.23 | 14-31 | 11-20 | TFIP11 |
| | | | | | | | | | 57-74 | 60-77 | |
| | | | | | | | | | | 102-107 | |
| Ntr2 | YKR022C | P36118 | 36.7 | 322 | 0.59 | 0.69 | 0.90 | 0.62 | | 1-0 | |
| | | | | | | | | | | 29-35 | |
| | | | | | | | | | | 46-59 | |
| | | | | | | | | | | 61-78 | 77-83 | |
| | | | | | | | | | | 96-113 | 96-114 | |
| | | | | | | | | | | 126-143 | 136-145 | |
| | | | | | | | | | | | 208-215 | |
| **CBP proteins** | | | | | | | | | | | |
| Sto1 | YMR125W | P34160 | 100.0 | 861 | 0.06 | 0.21 | 0.20 | 0.08 | 1-18 | | CBP80 |
| Cbc2 | YPL178W | Q08920 | 23.8 | 208 | 0.42 | 0.34 | 0.65 | 0.43 | 3-21 | 151-163 | CBP20 |
| | | | | | | | | | 180-197 | 186-196 | |
| **Other proteins** | | | | | | | | | | | |
| Prp5 | | A6ZLH6 | 96.4 | 849 | 0.35 | 0.44 | 0.49 | 0.46 | 1-18 | | |
| | | | | | | | | | 29-46 | 22-32 | |
| | | | | | | | | | 103-120 | 100-107 | |
| | | | | | | | | | | 117-122 | |
| | | | | | | | | | 145-162 | 156-171 | |
| | | | | | | | | | 661-678 | | |

| Yeast protein name | Gene name | UniProt ID PDB ID (residues) | MW (kDa) | Length | PONDR-FIT score | PONDR VLXT score | FoldIndex score | RONN score | MoRFs | AIBSs | Human protein name |
|---|---|---|---|---|---|---|---|---|---|---|---|
| | | | | | | | | | | 675–683 | |
| | | | | | | | | | 683–700 | 735–742 | |
| Usa1 | YML029W | Q03714 | 96.7 | 838 | 0.12 | 0.23 | 0.22 | 0.19 | | 359–365 | |
| | | | | | | | | | | 414–426 | |
| | | | | | | | | | | 828–838 | |
| Sqs1 | YNL224C | P53866 | 87.0 | 767 | 0.47 | 0.57 | 0.82 | 0.58 | | 1–39 | |
| | | | | | | | | | | 49–54 | |
| | | | | | | | | | 61–78 | 65–86 | |
| | | | | | | | | | 105–122 | 110–140 | |
| | | | | | | | | | 124–141 | | |
| | | | | | | | | | | 153–174 | |
| | | | | | | | | | | 202–213 | |
| | | | | | | | | | | 222–232 | |
| | | | | | | | | | | 237–251 | |
| | | | | | | | | | | 275–289 | |
| | | | | | | | | | 319–336 | 312–329 | |
| | | | | | | | | | | 350–358 | |
| | | | | | | | | | 506–523 | 505–512 | |
| | | | | | | | | | 542–559 | 540–545 | |
| | | | | | | | | | | 589–598 | |
| | | | | | | | | | 598–615 | | |
| | | | | | | | | | | 629–637 | |
| | | | | | | | | | | 675–680 | |
| | | | | | | | | | 698–715 | | |
| | | | | | | | | | | 747–756 | |
| Exo84 | YBR102C | P38261 2D2S (523–753) | 85.5 | 753 | 0.43 | 0.47 | 0.42 | 0.40 | 1–18 | 1–22 | |
| | | | | | | | | | 41–58 | 41–106 | |
| | | | | | | | | | | 129–152 | |
| | | | | | | | | | | 173–185 | |
| | | | | | | | | | 281–298 | | |
| | | | | | | | | | 462–479 | | |
| | | | | | | | | | 502–519 | 502–516 | |
| | | | | | | | | | 583–600 | | |
| Snu71 | | A6ZV04 | 71.4 | 620 | 0.45 | 0.36 | 0.68 | 0.48 | 270–287 | 275–295 | |
| | | | | | | | | | 316–333 | 315–327 | |

Table 1 (*continued*)

| Yeast protein name | Gene name | UniProt ID | PDB ID (residues) | MW (kDa) | Length | PONDR-FIT score | PONDR VLXT score | FoldIndex score | RONN score | MoRFs | AIBSs | Human protein name |
|---|---|---|---|---|---|---|---|---|---|---|---|---|
| | | | | | | | | | | 379-396 | 372-384 | |
| | | | | | | | | | | | 387-402 | |
| | | | | | | | | | | 407-424 | 405-428 | |
| | | | | | | | | | | | 450-474 | |
| | | | | | | | | | | | 495-507 | |
| | | | | | | | | | | | 566-577 | |
| Mud2 | YKL074C | P36084 | | 60.5 | 527 | 0.44 | 0.37 | 0.39 | 0.34 | 30-47 | 9-48 | |
| | | | | | | | | | | | 57-63 | |
| | | | | | | | | | | 88-105 | 66-113 | |
| | | | | | | | | | | 146-163 | 144-158 | |
| | | | | | | | | | | | 172-199 | |
| Msl5 | YLR116W | Q12186 | | 53.0 | 476 | 0.50 | 0.58 | 0.62 | 0.67 | | 1-7 | |
| | | | | | | | | | | | 62-79 | |
| | | | | | | | | | | | 12-136 | |
| | | | | | | | | | | | 163-168 | |
| | | | | | | | | | | | 215-220 | |
| | | | | | | | | | | | 233-239 | |
| | | | | | | | | | | 274-291 | 294-304 | |
| | | | | | | | | | | 300-317 | | |
| | | | | | | | | | | | 333-355 | |
| | | | | | | | | | | | 363-375 | |
| | | | | | | | | | | | 381-391 | |
| | | | | | | | | | | | 397-417 | |
| | | | | | | | | | | | 422-476 | |
| Thp3 | YPR045C | Q12049 | | 53.7 | 470 | 0.37 | 0.33 | 0.44 | 0.34 | | 1-9 | |
| | | | | | | | | | | 34-51 | 25-52 | |
| | | | | | | | | | | | 69-86 | |
| | | | | | | | | | | 121-138 | 88-133 | |
| | | | | | | | | | | 170-187 | 170-176 | |
| Sub2 | YDL084W | Q07478 | | 50.3 | 446 | 0.15 | 0.24 | 0.09 | 0.15 | | 3-27 | |
| | | | | | | | | | | | 56-80 | |
| Sub2 | | A6ZXP4 | | 50.3 | 446 | 0.15 | 0.24 | 0.09 | 0.15 | | 3-27 | |
| | | | | | | | | | | | 57-80 | |

Table 1 (*continued*)

| Yeast protein name | Gene name | UniProt ID PDB ID (residues) | MW (kDa) | Length | PONDR-FIT score | PONDR VLXT score | FoldIndex score | RONN score | MoRFs | AIBSs | Human protein name |
|---|---|---|---|---|---|---|---|---|---|---|---|
| Prp24 | YMR268C | P49960 2GHP (1–291) 2L9W (292–400) | 50.0 | 444 | 0.17 | 0.28 | 0.36 | 0.30 | 269–287 | 58–64 252–259 437–444 | |
| Csn12 | YJR084W | P47130 | 49.5 | 423 | 0.15 | 0.33 | 0.26 | 0.20 | 41–58 | | |
| Npl3 | YDR432W | Q01560 2JVO (114–201) 2JVR (193–282) | 45.4 | 414 | 0.67 | 0.63 | 0.79 | 0.72 | 71–88 120–137 262–279 301–318 386–403 | 1–52 55–90 120–136 141–146 155–164 261–267 291–310 323–342 358–368 377–405 | |
| Swt21 | YNL187W | P53873 | 40.3 | 357 | 0.04 | 0.19 | 0.16 | 0.14 | | 13–20 | |
| Brr1 | YPR057W | Q99177 | 39.9 | 341 | 0.14 | 0.27 | 0.49 | 0.23 | | | |
| Cus2 | YNL286W | P53830 | 32.3 | 285 | 0.37 | 0.41 | 0.68 | 0.42 | | 111–123 184–195 255–261 | |
| Mer1 | YNL210W | P16523 | 31.1 | 270 | 0.12 | 0.19 | 0.03 | 0.16 | | | |
| Spp381 | | A6ZL94 | 25.2 | 216 | 0.86 | 0.72 | 1.00 | 0.80 | 1–18 107–124 141–158 164–181 199–216 | 1–16 27–53 87–112 120–152 160–169 | |
| Rpl30 | YGL030W | P14120 1NMU (2–31) | 11.4 | 105 | 0.28 | 0.32 | 0.00 | 0.03 | | | |
| Hub1 | YNR032C-A | Q6Q546 3PLV (1–73) | 8.3 | 73 | 0.22 | 0.19 | 0.00 | 0.00 | | | |

**Notes.**

Proteins that have information on 3-D structures for entire proteins or some of their parts are highlighted in light blue. Highly disordered proteins selected for detailed functional and structural analysis are highlighted in light red. Highly disordered proteins selected for detailed functional analysis that have information on 3-D structures are highlighted in light pink.

in composition between a given set of proteins and a set of reference proteins (either a set of yeast spliceosomal proteins or a set of disordered proteins from DisProt database (*Vucetic et al., 2005*; *Sickmeier et al., 2007*)) was calculated for each amino acid residue. The fractional difference was calculated as $(C_X - C_{order})/C_{order}$, where $C_X$ is the content of a given amino acid in a query protein set, and $C_{order}$ is the corresponding content in a set of ordered proteins and plotted for each amino acid. In corresponding plots, the amino acids were arranged from the most order-promoting to the most disorder-promoting (*Radivojac et al., 2007*).

## Evaluation of the intrinsic disorder propensities

### Per residue disorder scores

The intrinsic disorder propensities of the spliceosomal proteins were evaluated by several different disorder predictors, such as PONDR® VLXT (*Dunker et al., 2001*), PONDR® VSL2 (*Peng et al., 2005*), PONDR® VL3 (*Peng et al., 2006*), FoldIndex (*Prilusky et al., 2005*), IUPred (*Dosztanyi et al., 2005a*), TopIDP (*Campen et al., 2008*), RONN (*Yang et al., 2005*), and PONDR® FIT (*Xue et al., 2010*). These predictors are briefly described below.

PONDR® VLXT applies various compositional probabilities and hydrophobic measures of amino acid as the input features of artificial neural networks for the prediction (*Romero et al., 2001*). PONDR® VLXT applies three different neural networks, one for each terminal region and one for the internal region of the sequence. Each neural network is trained by a specific dataset containing only the amino acid residues of that specific region. The final prediction result uses the individual predictors in their respective regions. The transition from one predictor to another is accomplished by computing the average scores of the two predictors for a short region of overlap at the boundary between the two regions. The input features of neural networks include selected compositions and profiles from the primary sequences. PONDR® VLXT may underestimate the occurrence of long disordered regions in proteins. Although it is no longer the most accurate predictor, it is very sensitive to the local compositional biases. Hence, this method has significant advantages in finding potential binding sites (*Oldfield et al., 2005a*; *Cheng et al., 2007*).

PONDR® VL3 employs ten neural networks and selects the final prediction by simple major voting. The input features of these predictors are various sequence profiles. This predictor has higher accuracy in predicting longer disordered regions (*Peng et al., 2006*).

PONDR® VSL2 is a combination of neural network predictors for both short and long disordered regions. A length limit of 30 residues divides short and long disordered regions. Each individual predictor is trained by the dataset containing sequences of that specific length. And the final prediction is a weighted average determined by a second layer predictor. PONDR® VSL2 applies not only the sequence profile, but also the result of sequence alignments from PSI-blast and secondary structure prediction from PHD and PSI-pred. This predictor is one the most accurate predictor in the PONDR family (*Peng et al., 2005*).

IUPred assumes that globular proteins have larger numbers of effective inter-residue interactions (negative free energy) than disordered proteins due to the different types of

amino acids involved in possible residue contacts. Based on this idea, a composition-based pair-wise interaction matrix was shown to give values similar to those obtained from a structure-based interaction matrix. Structured and disordered proteins were compared by this approach, with the structured proteins found to have a significantly lower free energy estimate, thus giving a means to predict whether a protein is structured or disordered using amino acid sequence as input (*Dosztanyi et al., 2005a*).

FoldIndex is a method developed from charge-hydropathy plots (*Uversky, Gillespie & Fink, 2000*) by rearranging the terms in the basic equation and by adding the technique of sliding windows (*Prilusky et al., 2005*). The charge-hydropathy plot was designed to determine if a protein is disordered or not. By applying a sliding window of 21 amino acids centered at a specific residue, the position of this segment on charge-hydrophobicity plot can be calculated, and the distance of this position away from the boundary line is taken as an indication whether the central residue is disordered or not (*Prilusky et al., 2005*).

TopIDP is a numerical scale giving the order–disorder propensity for each amino acid. This scale was determined by maximizing the differences in conditional probabilities for structured versus disordered regions of proteins for the central residues in windows of 21 residues (*Campen et al., 2008*).

PONDR® FIT (*Xue et al., 2010*) is a meta-predictor that combines six individual predictors, which are PONDR® VLXT (*Romero et al., 2001*), PONDR® VSL2 (*Peng et al., 2005*), PONDR® VL3 (*Peng et al., 2006*), FoldIndex (*Prilusky et al., 2005*), IUPred (*Dosztanyi et al., 2005a*), TopIDP (*Campen et al., 2008*). This meta-predictor is moderately more accurate than each of the component predictors.

RONN is the regional order neural network software that applies the "biobasis function neural network" pattern recognition algorithm for the detection of natively disordered regions in proteins. It predicts disordered structures based on the sequence alignments (*Yang et al., 2005*).

*Binary disorder predictions.* Cumulative distribution function curves or CDF curves (*Oldfield et al., 2005b*) were generated for each dataset using PONDR® FIT scores for each of the spliceosomal proteins. CDF analysis discriminates between order and disorder by means of a boundary value (*Xue et al., 2009*). This value can be interpreted as a measure of proportion of residues with low and high disorder predictions. Additionally, charge-hydropathy distributions (CH-plots) were also analyzed for these proteins using methods as described in *Uversky, Gillespie & Fink (2000)*.

*α-MoRF predictions.* The predictor of α-helix forming Molecular Recognition Features, α-MoRF, is based on observations that predictions of order in otherwise highly disordered proteins corresponds to protein regions that mediate interaction with other proteins or nucleic acids. This predictor focuses on short binding regions within long regions of disorder that are likely to form helical structure upon binding (*Oldfield et al., 2005a*). It uses a stacked architecture, where PONDR® VLXT is used to identify short predictions of order within long predictions of disorder and then a second level predictor determines whether the order prediction is likely to be a binding site based on attributes of both the predicted ordered region and the predicted surrounding disordered region. An α-MoRF

prediction indicates the presence of a relatively short (20 residues), loosely structured helical region within a largely disordered sequence (*Oldfield et al., 2005a*; *Cheng et al., 2007*). Such regions gain functionality upon a disorder-to-order transition induced by binding to partners (*Mohan et al., 2006*; *Vacic et al., 2007b*).

*ANCHOR analysis.* In addition to MoRF identifiers, potential binding sites in disordered regions can be identified by the ANCHOR algorithm (*Dosztanyi, Meszaros & Simon, 2009*; *Meszaros, Simon & Dosztanyi, 2009*). This approach relies on the pairwise energy estimation approach developed for the general disorder prediction method IUPred (*Dosztanyi et al., 2005a*; *Dosztanyi et al., 2005b*), being based on the hypothesis that long regions of disorder contain localized potential binding sites that cannot form enough favorable intrachain interactions to fold on their own, but are likely to gain stabilizing energy by interacting with a globular protein partner (*Dosztanyi, Meszaros & Simon, 2009*; *Meszaros, Simon & Dosztanyi, 2009*). Here we are using the term ANCHOR-indicated binding site (AIBS) to identify a region of a protein suggested by the ANCHOR algorithm to have significant potential to be a binding site for an appropriate but typically unidentified partner protein.

## Structural and functional annotation of selected proteins

We selected the 24 most disordered spliceosomal proteins according to an average between the disorder scores calculated by different predictors for more focused analysis of their structures, disorder propensities, and functions. In addition to the level of predicted intrinsic disorder, these proteins were chosen to represent all the major components and complexes comprising the yeast spliceosome. These proteins were researched for their function, structures, location within the spliceosome, etc. This information was obtained from the UniProtKB, and validated through the literature search.

## RESULTS AND DISCUSSION

### Evaluation of the abundance of intrinsic disorder in yeast spliceosomal proteins

To test for a correlation between the yeast spliceosomal proteins and intrinsic disorder, a dataset of 109 proteins associated with the yeast spliceosome was extracted from UniProt as described in Materials and Methods. Next, this set of proteins was analyzed using a broad spectrum of computational tools for the evaluation of intrinsic disorder in proteins. Results of this analysis are discussed below.

*Analysis of the compositional biases.* Since the amino acid sequences and compositions of IDPs and IDPRs are significantly different from those of ordered proteins and folded domains, a simple analysis of the amino acid composition biases can provide interesting information on the nature of a protein. For example, the amino acid compositions of extended IDPs (i.e., those disordered proteins that do not have almost any residual structure and behave as native coils and native pre-molten globules (*Dunker et al., 2001*; *Uversky, 2002a*; *Uversky, 2002b*; *Uversky, 2003*; *Uversky & Dunker, 2010*)) are characterized

by low mean hydropathy and high mean net charge, which define the highly unstructured and extended state of these proteins, since high net charge leads to strong electrostatic repulsion, and low hydropathy prevents efficient compaction (*Uversky, Gillespie & Fink, 2000*). Overall, IDPs/IDPRs are known to be significantly depleted in so-called order-promoting amino acids, C, W, I, Y, F, L, H, V, and N, and substantially enriched in disorder-promoting residues, A, G, R, T, S, K, Q, E, and P (*Dunker et al., 2001*; *Romero et al., 2001*; *Williams et al., 2001*; *Radivojac et al., 2007*; *Vacic et al., 2007a*). Therefore, the evaluation of the amino acid biases in a set of proteins can be used as a fast and informative way to evaluate their intrinsically disordered nature. This analysis can be done using a computational tool, Composition Profiler (*Vacic et al., 2007a*), which is based on the calculation of a normalized composition of a given protein or protein dataset in the $(C_x - C_{order})/C_{order}$ form, where $C_x$ is a content of a given residue in a query dataset, and $C_{order}$ is the corresponding value for the set of ordered proteins from PDB Select 25 (*Berman et al., 2000*).

Results of this analysis are shown in Fig. 1A, which illustrates that, in comparison with typical ordered proteins, yeast spliceosomal proteins are moderately depleted in some order-promoting residues (e.g., C, W, Y, F, H, and V, see orange bars in Fig. 1A) and are moderately enriched in some major disorder-promoting residues (e.g., D, K, Q, S and E). On the other hand, some order-promoting residues (I, L and M) are rather common in these proteins, whereas some disorder-promoting residues (G, A, and P) are clearly underrepresented in yeast spliceosome. Both depletion in major order-promoting residues and enrichment in major disorder-promoting residues suggest that the yeast spliceosomal proteins might contain multiple signatures characteristic for the disordered proteins.

*Abundance of long disordered regions in yeast spliceosomal proteins.* Previous study revealed that intrinsic disorder is very abundant in signaling proteins, and this abundance can be evaluated by estimating the fraction of proteins with long disordered regions (*Iakoucheva et al., 2002*). In fact, the application of PONDR® VLXT (*Romero et al., 2001*) showed that 66% of cell-signaling proteins contain predicted regions of disorder of 30 residues or longer (*Iakoucheva et al., 2002*). Therefore, we applied similar approach and systematically analyzed the intrinsic disorder tendencies in four protein datasets: (1) 109 yeast spliceosomal proteins (spliceosome); (2) 2,329 signaling proteins collected by the Alliance for Cellular Signaling (AfCS); (3) 53,630 eukaryotic proteins from UniProt (EU_UP); and (4) a set of 1,138 non-homologous protein segments with well-defined 3-D structure from the Protein Data Bank Select 25 (O_PDB_S25). Figure 1B illustrates that intrinsic disorder is prevalent in the yeast spliceosomal proteins, being comparable with the prevalence observed for signaling and eukaryotic proteins. In fact, the percentages of proteins with 30 or more consecutive residues predicted to be disordered were 53% for the spliceosomal proteins, 66% for AfCS, 47% for EU_SW, and 13% for O_PDB_S25. In other words, the fraction of yeast spliceosomal proteins with long regions of predicted disorder is 4-fold higher than that of non-homologous ordered proteins from PDB (*Iakoucheva et al., 2002*), being also a bit higher than the corresponding fraction in eukaryotic proteins.

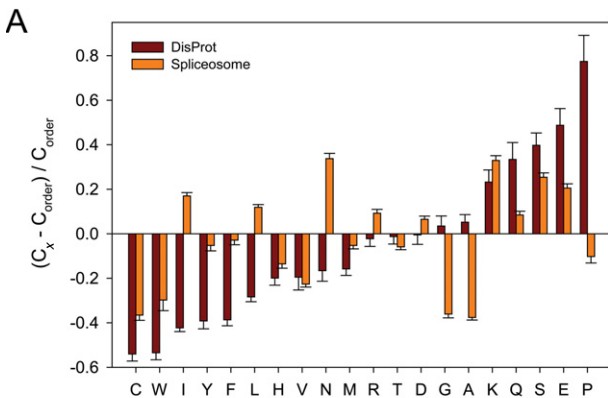

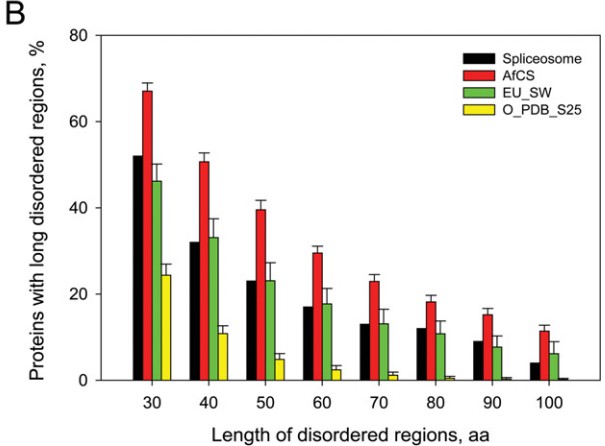

**Figure 1 Evaluation of abundance of intrinsic disorder in the yeast spliceosome.** A. Fractional difference in the amino acid composition between the different yeast spliceosomal proteins and a set of completely ordered proteins calculated for each amino acid residue (compositional profiles). The fractional difference was evaluated as $(C_x - C_{order})/C_{order}$, where $C_x$ is the content of a given amino acid in a query set, and $C_{order}$ is the corresponding content in the dataset of fully ordered proteins. Composition profile of typical IDPs from the DisProt database is shown for comparison (black bars). Positive bars correspond to residues found more abundantly in histones, whereas negative bars show residues, in which histones are depleted. Amino acid types are ranked according to their increasing disorder-promoting potential (*Radivojac et al., 2007*). B. Abundance of predicted long disordered regions in yeast spliceosomal proteins (black bars) in comparison with long disordered regions in 2,329 proteins involved in cellular signaling (AfCS, red bars), 53,630 eukaryotic proteins from SWISS-PROT (EU_SW, green bars), and 1,138 sequences corresponding to ordered parts of proteins from PDB Seect 25 (O_PDB_S25, yellow bars).

*Disorder propensity of yeast spliceosomal proteins studied by the binary disorder predictors.* Sequences of the 109 yeast spliceosomal proteins were used to predict whether these proteins are likely to be mostly disordered using two binary predictors of intrinsic disorder: charge-hydropathy plot (CH-plot) (*Uversky, Gillespie & Fink, 2000*; *Oldfield et al., 2005b*) and cumulative distribution function analysis (CDF) (*Oldfield et al., 2005b*). Both these methods perform binary classification of whole proteins as either mostly disordered or mostly ordered, where mostly ordered indicates proteins that contain more ordered residues than disordered residues and mostly disordered indicates proteins that contain more disordered residues than ordered residues (*Oldfield et al., 2005b*).

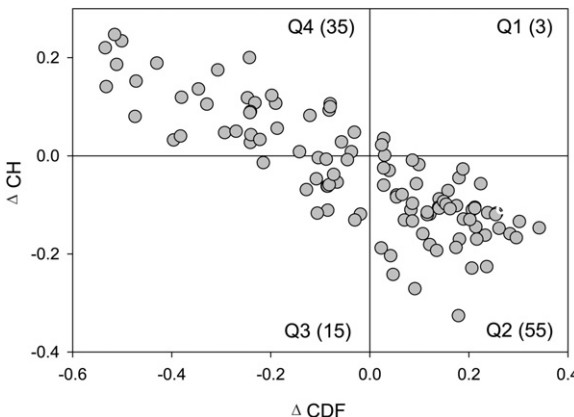

**Figure 2  CH-CDF analysis of the yeast spliceosomal proteins.** Here, the coordinates of each point were calculated as a distance of the corresponding protein in the CH-plot from the boundary (Y-coordinate) and an average distance of the respective CDF curve from the CDF boundary (X-coordinate). The four quadrants correspond to the following predictions: Q1, proteins predicted to be disordered by CH-plots, but ordered by CDFs; Q2, ordered proteins; Q3, proteins predicted to be disordered by CDFs, but compact by CH-plots (i.e., putative molten globules or mixed proteins); Q4, proteins predicted to be disordered by both methods (i.e., proteins with extended disorder).

Figure 2 represents the results of the combined CH-CDF analysis of the spliceosomal proteins and shows that ∼50% of these proteins are mostly disordered. In this plot, the coordinates of each spot are calculated as a distance of the corresponding protein in the CH-plot (charge-hydropathy plot) from the boundary (Y-coordinate) and an average distance of the respective cumulative distribution function (CDF) curve from the CDF boundary (X-coordinate) (*Mohan et al., 2008*; *Xue et al., 2009*; *Huang et al., 2012*). The primary difference between these two binary predictors (i.e., predictors which evaluate the predisposition of a given protein to be ordered or disordered as a whole) is that the CH-plot is a linear classifier that takes into account only two parameters of the particular sequence (charge and hydropathy), whereas CDF analysis is dependent on the output of the PONDR® predictor, a nonlinear classifier, which was trained to distinguish order and disorder based on a significantly larger feature space. According to these methodological differences, CH-plot analysis is predisposed to discriminate proteins with substantial amount of extended disorder (random coils and pre-"molten globules") from proteins with compact conformations ("molten globule"-like and rigid well-structured proteins). On the other hand, PONDR-based CDF analysis may discriminate all disordered conformations, including molten globules and mixed proteins containing both disordered and ordered regions, from rigid well-folded proteins. Therefore, this discrepancy in the disorder prediction by CDF and CH-plot provides a computational tool to discriminate proteins with extended disorder from potential molten globules and mixed proteins.

Positive and negative Y values in Fig. 2 correspond to proteins predicted within CH-plot analysis to be natively unfolded or compact, respectively. On the other hand, positive and negative X values are attributed to proteins predicted within the CDF analysis to be ordered or intrinsically disordered, respectively. Thus, the resultant quadrants of CDF-CH phase space correspond to the following expectations: Q1, proteins predicted to be disordered

by CH-plots, but ordered by CDFs; Q2, ordered proteins; Q3, proteins predicted to be disordered by CDFs, but compact by CH-plots (i.e., putative molten globules or mixed proteins); Q4, proteins predicted to be disordered by both methods (i.e., proteins with extended disorder).

Figure 2 shows that ~50% of the yeast spliceosomal proteins are predicted to be disordered as a whole, with 33% and 13.8% of them being found in quadrants Q4 and Q3, respectively, and are therefore expected to behave as native coils or native pre-molten globules or native molten globules or mixed proteins in their unbound states. The fact that 46.7% of the spliceosomal proteins are expected to be mostly disordered (being located within quadrants Q3 and Q4) is a very important observation since this value noticeably exceeds the corresponding value evaluated for the yeast proteins in general (13.3%) (*Mohan et al., 2008*).

*Combined analysis of intrinsic disorder propensity by several computational tools.* It was emphasized that the combined analysis of the intrinsic disorder propensity by several computational tools (especially by tools that utilizes different attributes) provides additional advantages (*Ferron et al., 2006*; *Bourhis, Canard & Longhi, 2007*; *He et al., 2009*), allowing, for example, better visualization of the differences between the various protein groups (*Uversky et al., 2006*). Figure 3A illustrates the power of this approach and represents a plot where disorder contents in the yeast spliceosomal proteins were evaluated by PONDR-FIT, which is a meta-predictor that provides more accurate disorder content predictions when compared to several other recent disorder predictors (*Xue et al., 2010*), and PONDR® VLXT (*Romero et al., 2001*), which is no longer the most accurate predictor, but is very sensitive to the local compositional biases and is capable of identifying potential molecular interaction motifs (*Oldfield et al., 2005a*; *Cheng et al., 2007*). In our analysis, we used two arbitrary cutoffs for the levels of intrinsic disorder to classify proteins as highly ordered ([IDP score] < 10%), moderately disordered (30% > [IDP score] > 10%) and highly disordered ([IDP score] > 30%) (*Rajagopalan et al., 2011*). According to this separation, just 9% of the proteins were predicted to be highly ordered by PONDR-FIT, with 48% and 52% of proteins classified as moderately and highly disordered, respectively (see Fig. 3A). This grouping suggests that most of the proteins in the spliceosome are intrinsically disordered.

Since PONDR-FIT is a metapredictor that includes PONDR® VLXT as one of its components, a linear relationship between the results of these two predictors was expected. Therefore, we used a more complex analysis, where the outputs of three truly independent approaches were compared. Figure 3B represents the results of this analysis and shows the 3D disorder distribution plot, where the outputs of PONDR-FIT, RONN and FoldIndex are used as three dimensions. This representation clearly shows that the outputs of three very different computational tools (see Materials and Methods for the description of these tools) are generally agree with each other, since the points corresponding to the different spliceosomal proteins are mostly located on the diagonal of the FIT-RONN-FoldIndex space.

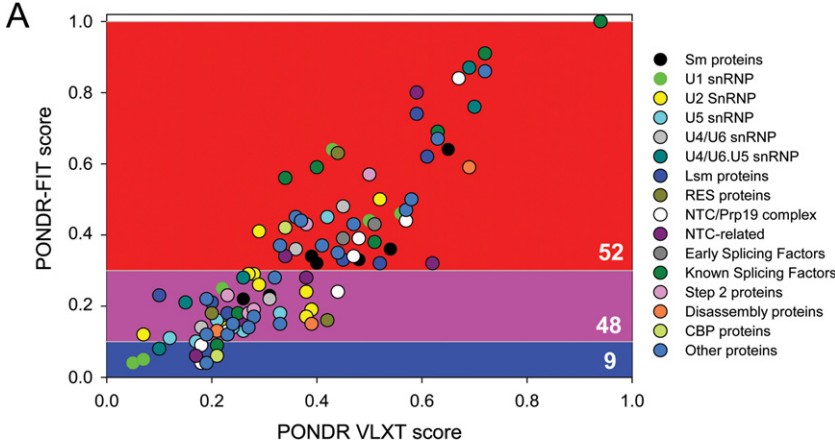

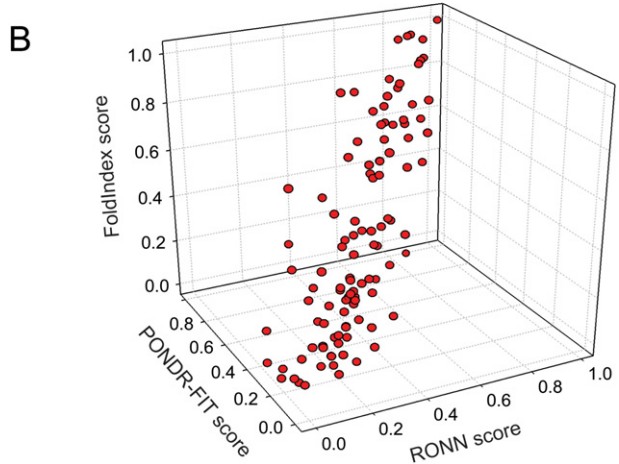

**Figure 3 Combined analysis of intrinsic disorder propensities of the yeast spliceosomal proteins using the outputs of different disorder prediction tools.** A. PONDR-FIT vs. PONDR® VLXT plot representing the correlation between the disorder content evaluated by PONDR-FIT (*y*-axis) (*Xue et al., 2010*) and by PONDR® VLXT (*x*-axis) (*Romero et al., 2001*). Two arbitrary cutoffs for the levels of intrinsic disorder were used to classify proteins as highly ordered ([IDP score] < 10%, blue field), moderately disordered (30% > [IDP score] > 10%, pink field) and highly disordered ([IDP score] > 30%, red field) (*Rajagopalan et al., 2011*). Color coding of spliceosomal proteins reflects their relation to different components and complexes. B. 3D disorder distribution plot representing the PONDR-FIT vs. RONN vs. FoldIndex dependence.

## Functions of IDPs and IDPRs in yeast spliceosome

*Distribution of IDPs in different components of the yeast spliceosome.* The spliceosome of any organism is a protein-rich molecular machine (*Fabrizio et al., 2009*). In fact, the major spliceosome contains five uridine-rich small nuclear RNAs (U1, U2, U4, U5, and U6) that are responsible for the catalysis of the pre-mRNA splicing and that are assisted by a wide array of proteins, number of which ranges from ∼100 (in yeast) to more than 200 (in metazoan). Depending on their involvement in the formation of snRNPs, spliceosomal proteins can be grouped into two major categories, proteins associated with snRNPs and non-snRNP spliceosomal proteins. Since the spliceosome is a highly dynamic machine,
the number of the spliceosome's protein complement varies substantially from one stage of the splicing cycle to another (*Fabrizio et al., 2009*). For example, the transition from the complex B to complex C is accompanied not only by the dissociation of U1 and U4 snRNAs from the spliceosme but by the dramatic perturbation in the protein composition, where ∼35 proteins are removed and new 12 spliceosomal proteins are added to the complex (*Bessonov et al., 2008*; *Fabrizio et al., 2009*).

Figure 4 illustrates compositional changes that take place at the different stages of the spliceosome assembly and action and shows the protein compositions of the yeast B, B$^{act}$, and C complexes determined by mass spectrometry (*Fabrizio et al., 2009*). Here, the involved proteins are color coded according to their intrinsic disorder content evaluated by PONDR-FIT, with highly ordered (ID score < 10%), moderately disordered (30% > ID score > 10%) and highly disordered proteins (DP score > 30%) being shown as blue, pink and red bars, respectively. Details of this analysis are further summarized in Table 1, which in addition to the major structural properties of the spliceosomal proteins lists their intrinsic disorder scores evaluated by four different disorder predictors.

*Predictions of potential disorder-based binding sites, α-MoRFs.* Often, intrinsically disordered regions in proteins are involved in protein–protein interactions and molecular recognitions (*Dunker et al., 2001*; *Dunker et al., 2002*; *Dunker, Brown & Obradovic, 2002*; *Tompa, 2002*; *Daughdrill et al., 2005*; *Dunker et al., 2005*; *Uversky, Oldfield & Dunker , 2005*; *Radivojac et al., 2007*; *Dunker et al., 2008a*; *Dunker & Uversky, 2008*; *Uversky & Dunker, 2010*; *Uversky, 2011*; *Uversky, 2012*). Many flexible proteins or regions undergo disorder-to-order transitions upon binding, which is crucial for recognition, regulation, and signaling (*Wright & Dyson, 1999*; *Uversky, Gillespie & Fink, 2000*; *Dunker et al., 2001*; *Dyson & Wright, 2002*; *Dyson & Wright, 2005*; *Oldfield et al., 2005a*; *Mohan et al., 2006*; *Vacic et al., 2007b*). A correlation has been established between the specific pattern in the PONDR® VLXT curve and the ability of a given short disordered regions to undergo disorder-to-order transitions on binding (*Garner et al., 1999*). Based on these specific features in the protein's disorder profile and a set of attributes of both the predicted ordered region and the predicted surrounding disordered region specific predictors of α-helix forming Molecular Recognition Features, α-MoRFs, were developed (*Oldfield et al., 2005a*; *Cheng et al., 2007*). An α-MoRF prediction indicates the presence of a relatively short, loosely structured helical region within a largely disordered sequence (*Oldfield et al., 2005a*). Such regions gain functionality upon a disorder-to-order transition induced by binding to partners (*Mohan et al., 2006*; *Vacic et al., 2007b*).

Application of the α-MoRF predictors reveals that molecular recognition features are highly abundant in yeast spliceosomal proteins, and Table 1 shows that ∼61% spliceosomal proteins contain α-MoRFs. This value is almost 3-fold larger than the corresponding value evaluated for the yeast proteins in general (21.1%) (*Mohan et al., 2008*). On average, each protein associated with east spliceosome contains 1.75 α-MoRFs, which is noticeably larger than 0.39 α-MoRFs per yeast protein in general (*Mohan et al., 2008*). Also, on average, each protein in the yeast proteome that was predicted to possess α-MoRFs was shown to have 1.84 molecular recognition features (*Mohan et al., 2008*). In spliceosome,

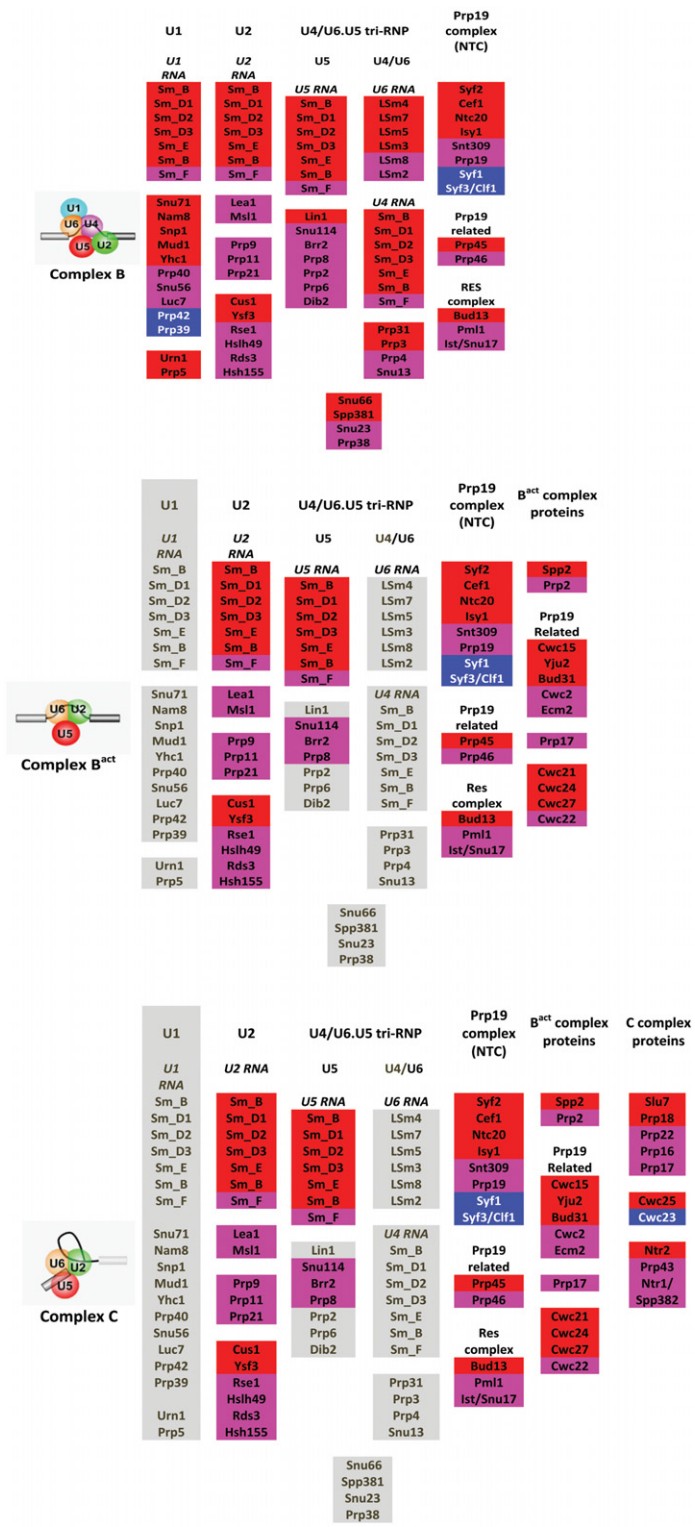

**Figure 4  Compositional changes taking place at the different stages of the spliceosome assembly (*Fabrizio et al., 2009*).** The proteins are color coded according to their intrinsic disorder content evaluated by PONDR-FIT, with highly ordered ([IDP score] < 10%), (continued on next page...)

> **Figure 4 (...continued)**
> moderately disordered (30% > [IDP score] > 10%) and highly disordered proteins ([IDP score] > 30%)
> being shown as blue, pink and red bars, respectively. Gray bars correspond to proteins that are present in
> the complex B only and are not seeing at subsequent stages; i.e., excluded from complexes B$^{act}$ and C.

MoRF-possessing proteins contain 2.58 $\alpha$-MoRFs per protein (see Table 1). Importantly, some long, highly disordered spliceosomal proteins have multiple predicted $\alpha$-MoRF regions (Table 1) that may potentially serve as binding sites for multiple proteins. For example, Snu66 (687 amino acid residues) has 11 predicted $\alpha$-MoRFs, whereas there are 7, 6, and 5 predicted $\alpha$-MoRFs in Prp3 (469 amino acid residues), Spp381 (191 amino acids), and Yju2 (278 residues) respectively. All this suggests that the spliceosomal proteins are extremely enriched in disorder-based binding sites and therefore are involved in extensive interaction networks.

*Predictions of potential disorder-based binding sites, AIBSs.* In addition to the PONDR-based MoRF identifiers which find disorder-driven binding sites using the peculiarities of predicted disorder propensity distribution within a protein sequence, potential binding sites in disordered regions can be identified by the ANCHOR algorithm (*Dosztanyi, Meszaros & Simon, 2009*; *Meszaros, Simon & Dosztanyi, 2009*). In order to predict disordered binding regions, ANCHOR identifies segments (ANCHOR-identified binding sites, AIBSs) that reside in disordered regions, cannot form enough favorable intrachain interactions to fold on their own, and are likely to gain stabilizing energy by interacting with a globular protein partner (*Dosztanyi, Meszaros & Simon, 2009*; *Meszaros, Simon & Dosztanyi, 2009*). Therefore, methodologically and logistically, ANCHOR is very different from the MoRF identifiers.

Table 1 represents the results of the ANCHOR-based analysis of the yeast spliceosomal proteins and shows AIBSs are very common in these proteins. In fact, of the 109 yeast spliceosomal proteins analyzed in this study 77 contained at least one AIRS. Therefore, AIBSs were found in ∼71% yeast spliceosomal proteins. Analysis data shown in Table 1 shows that there is generally a good agreement between the results of binding sites predictions by MoRF identifiers and ANCHOR. For proteins containing disorder-based binding sites, there are typically more AIBSs than MoRFs. This is an expected result since MoRF identifiers are designed to find disordered regions that fold into $\alpha$-helices at interaction with the binding partners, whereas ANCHOR is a more general method which is not biased toward any type of the protein secondary structure in the bound state.

## Structures and functions of some highly disordered spliceosomal proteins

Spliceosome assembly is a multistep process that involves sequential binding of snRNPs to the pre-mRNA in an order of U1, U2, then U4/U6 and U5 as a preformed tri-snRNP particle. A subsequent conformational rearrangement results in dissociation of U1 and U4, accompanied by new base pair formation between U2 and U6 and between U6 and the 5' splice site, leading to the formation of the active spliceosome on which the

catalytic reactions take place (*Chen et al., 2001*). snRNAs (which are the central structural and functional units of spliceosomal snRNPs) have important roles in recognition and alignment of splice sites mediated through base pair interactions between snRNAs and the intron sequences during spliceosome assembly (*Chen et al., 2001*). Furthermore, it is believed that snRNAs of these snRNPs act as ribozymes, being responsible for the catalysis of the intron excision (*Abelson, 2008*; *Pyle, 2008*; *Fabrizio et al., 2009*). However, all the steps related to the spliceosome assembly and actions are known to be accompanied by the dramatic rearrangements of the spliceosomal protein composition. This suggests that protein-based interactions are crucial for the spliceosome function.

From the 109 proteins studies in this work, 24 highly disordered spliceosomal proteins (Cwc21, Ntc20, Isy1/Ntc30, Prp45, Snu66, Cwc15, Spp381, Syf2, Cwc26, Slu7, Yju2/Cwc16, Ntr2, Npl3, Spp2, Bud31, SmB, Yhc1, Cus1, Lin1, Prp3, Lsm4, Prp5, Cbc2, and Msl5) were selected for more focused analysis of their structures, disorder propensities, functions, post-translational modifications, and the presence or lack of 3-D structures solved for the entire proteins or for some of their parts. In addition to the level of predicted intrinsic disorder, these proteins were chosen to represent all the major components of the yeast spliceosome.

*Pre-mRNA-splicing factor Cwc21 or complexed with Cef1 protein 21 (UniProt ID: Q03375).* Cwc21 protein is a part of the U2-type spliceosome complex and its putative role is the stabilization of the catalytic site or the position of RNA substrate during the splicing process. In *S. cerevisiae*, Cwc21 binds to two key splicing factors, namely, Prp8 and Snu114, and docks directly to U5 snRNP. It was demonstrated that SRm300, the only SR-related protein known to be at the core of human catalytic spliceosomes, is a functional ortholog of Cwc21, which also interacts directly with Prp8 and Snu114 (*Grainger et al., 2009*). Thus, the function of Cwc21 is likely to be conserved from yeast to humans. Cwc21 also shows affinity for the protein Isy1, a splicing fidelity factor, indicating that, even though it is not an essential protein for the function and formation of the spliceosome (*Hogg, McGrail & O'Keefe, 2010*), it is required for the correct splicing (*Khanna et al., 2009*). Cwc21 is a small highly basic protein (pI 9.67, 135 residues), that interacts with Prp8 via SCwid domain (53-97 region) and Snu114 (via C-terminus) (*Grainger et al., 2009*). Figure 5A and Table 1 show that Cwc21 is predicted to be highly disordered by PONDR-FIT and possesses two $\alpha$-MoRFs, one of which partially overlaps with the experimentally established Prp8 and Snu114 binding sites.

*Pre-mRNA-splicing factor Ntc20 or Prp19-associated complex protein 20 (UniProt ID:P38302) and pre-mRNA-splicing factor Isy1 or Ntc30 (UniProt ID:P21374).* The yeast *S. cerevisiae* Prp19 protein is an essential splicing factor and an important spliceosomal component. It is not tightly associated with small nuclear RNAs (snRNAs) but represents a core of a protein complex (NTC complex) consisting of at least eight proteins. Two of this NTC/Prp19-associated complex, proteins Ntc30 and Ntc20, associate to the spliceosome to mediate conformational rearrangement or to stabilize the structure of the spliceosome after U4 snRNA dissociation, which leads to spliceosome maturation (*Ben-Yehuda et al., 2000*; *Chen et al., 2001*; *Chen et al., 2002*; *Chan et al., 2003*). Null *NTC30*

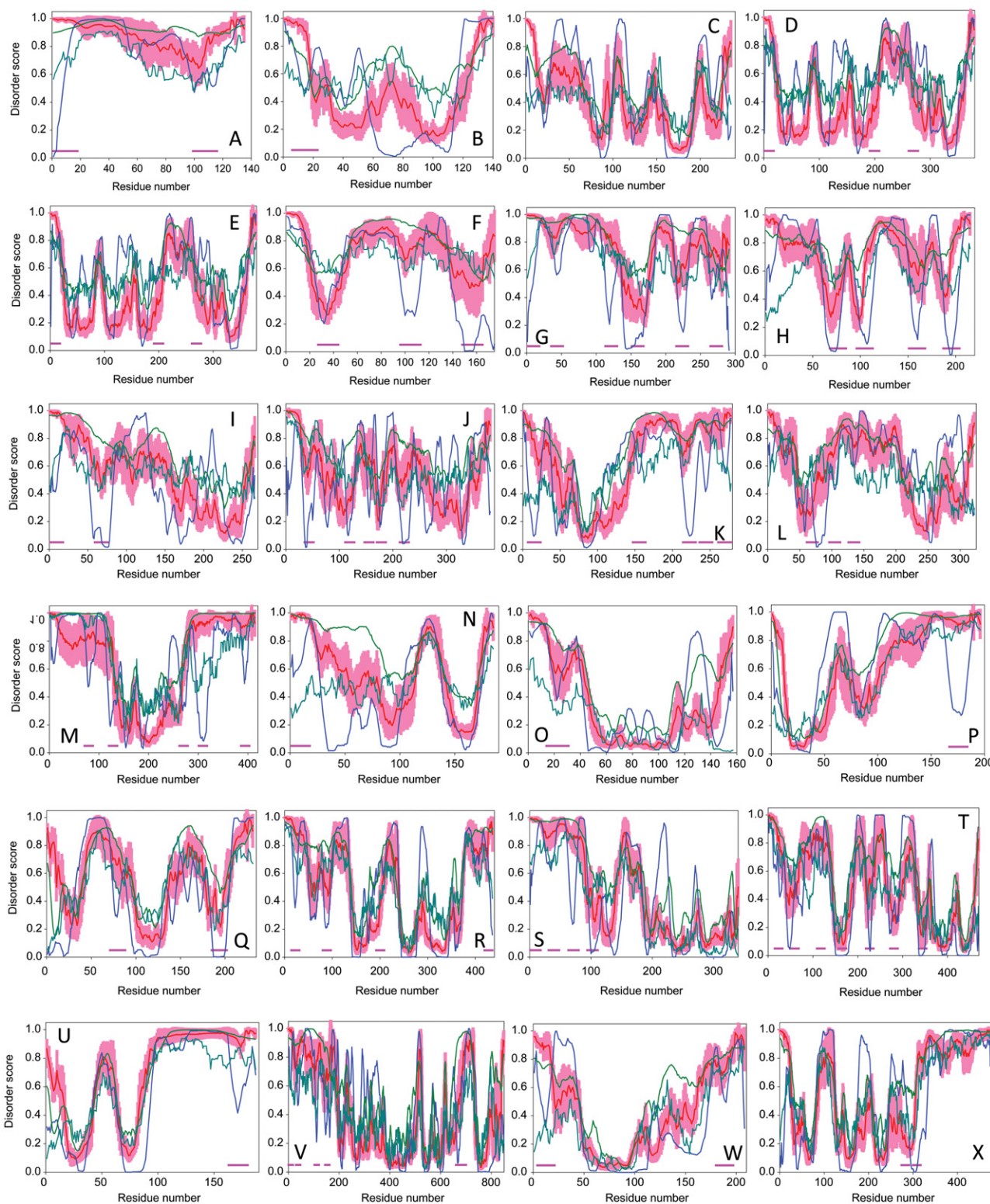

**Figure 5 Analysis of disorder distribution in illustrative spliceosomal proteins.** A. Cwc21; B. Ntc20; C. Isy1/Ntc30; D. Prp45; E. Snu66; F. Cwc15; G. Spp381; H. Syf2; I. Cwc26; J. Slu7; K. Yju2/Cwc16; L. Ntr2; M, Npl3; N. Spp2; O. Bud31; P. SmB; Q. Yhc1; (continued on next page...)

**Figure 5 (...continued)**

R. Cus1; S. Lin1; T. Prp3; U. LSm4; V. Prp5; W. Cbc2; and X. Msl5. For all these proteins, the disorder propensity was evaluated by PONDR® FIT (red curves); PONDR® VLXT (blue curves); PONDR® VSL2B (dark green curves); and IUPred (dark cyan curves). Shadow around PONDR® FIT curves represents distribution of statistical errors. Bold pink lines correspond to the predicted $\alpha$-MoRFs.

or *NTC20* mutants do not show obvious growth phenotype. However, simultaneous deletion of both genes impaired yeast growth resulting in accumulation of precursor mRNA, suggesting that Ntc30 and Ntc20 are auxiliary splicing factors the functions of which may be related to the modulation of the NTC complex function required for stable association of U5 and U6 with the spliceosome after U4 is dissociated (*Chen et al., 2001*).

Ntc20 is a small acidic protein (pI 5.93, 140 residues), whereas Ntc30 (also known as Isy1) is an average size basic protein (pI 9.35, 235 residues). Ntc20 interacts with Cef1, Clf1, Isy1/Ntc30, Prp46, and Syf1 proteins, which are components of the NTC complex (*Ben-Yehuda et al., 2000*; *Chen et al., 2001*). Exact locations of the potential binding sites are known, but Ntc20 was shown to be phosphorylated at position Ser139 (*Albuquerque et al., 2008*). Ntc30 interacts with Cef1, Cwc2, Clf1, and Syf1 (*Dix et al., 1999*; *Ben-Yehuda et al., 2000*; *Chen et al., 2001*). Both Ntc30 and Ntc20 are predicted to contain significant amount of disorder (see Table 1 and Figs. 5B and 5C).

*Pre-mRNA-processing protein 45, Prp45 (UniProt ID: P28004).* Prp45 is the yeast ortholog of the human Snw1/Skip transcription co-regulator, which regulates transcription elongation and alternative splicing, and was shown to genetically interacts with alleles of the NTC family members Syf1, Clf1/Syf3, Ntc20, and Cef1, and the second step splicing factors Slu7, Prp17, Prp18, and Prp22 (*Gahura et al., 2009*). Prp45 was suggested to contribute to splicing efficiency of substrates non-conforming to the consensus via its interaction with the second step-proofreading helicase Prp22 (*Gahura et al., 2009*). The functional equivalency of Prp45 and Skip was verified by the rescue of the Prp45 deleted lethal mutants by the insertion of a functional copy of the Skip gene in yeast (*Figueroa & Hayman, 2004*). It was shown that Prp45 interacts with Prp46 in vitro, demonstrating that these proteins are spliceosome-associated throughout the splicing process and both are essential for pre-mRNA splicing (*Albers et al., 2003*). Prp45 is known to be associated with the spliceosome throughout the splicing reactions, until after the second catalytic step (*Martinkova et al., 2002*; *Albers et al., 2003*). Prp45 is a basic protein (pI 9.15) that consists of 379 residues. It is predicted to contain significant amount of intrinsic disorder and contain three $\alpha$-MoRFs (see Table 1 and Fig. 5D).

*66 kDa U4/U6.U5 small nuclear ribonucleoprotein component (UniProt ID: Q12420).* The yeast U4/U6.U5 tri-snRNP is a 25S snRNP particle similar in size, composition, and morphology to its counterpart in human cells (*Stevens & Abelson, 1999*). Stevens and Abelson purified this complex and showed that there are at least 24 proteins stably associated with this particle. In addition to the seven canonical core Sm proteins, there are a set of U6 snRNP specific Sm proteins, eight previously described U4/U6.U5 snRNP proteins, and four novel proteins. Two of the novel proteins have likely RNA binding properties, one has been implicated in the cell cycle, and one has no identifiable sequence homologues

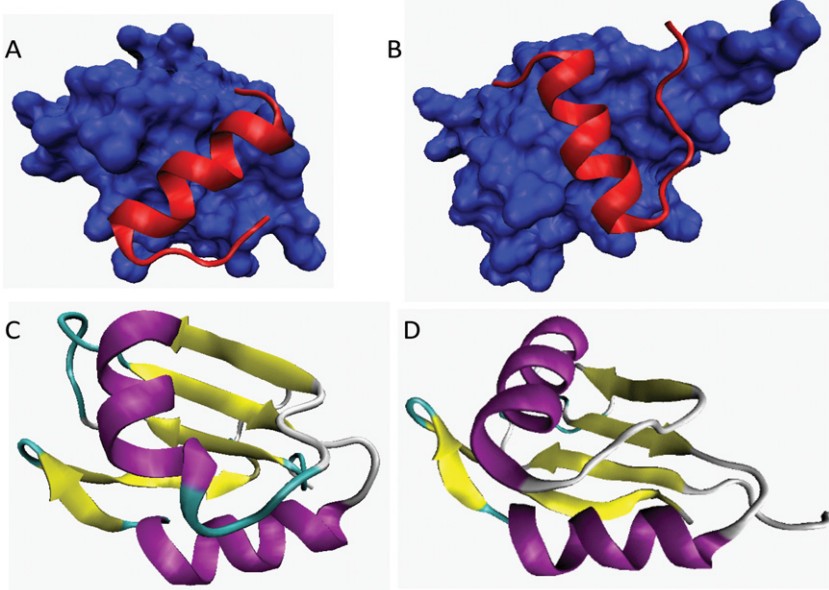

**Figure 6 3D-structures of fragments and domains of two highly disordered spliceosomal proteins, Snu66 (plots A and B) and Npl13 (plots C and D).** Structure visualizations were done with the VMD 1.8.7 (*Humphrey, Dalke & Schulten, 1996*). A. Crystal structure of Hub1 (shown as blue surface) in a complex with HIND-I element of Snu66 (residues 1-31, shown as the red ribbon) (PDB ID: 3PLU). B. Crystal structure of Hub1 (shown as blue surface) in a complex with HIND-II element of Snu66 (residues 32-62, shown as the red ribbon) (PDB ID: 3PLV). C. Solution structure the first RRM domain of the nucleolar protein Npl3 (residues 114-201) determined by NMR (PDB ID: 2JVO). D. Solution structure the second RRM domain of the nucleolar protein Npl3 (residues 193-282) determined by NMR (PDB ID: 2JVR).

or functional motifs. One of the proteins associated with U4/U6.U5 tri-snRNP is Snu66, which is required for pre-mRNA splicing (*van Nues & Beggs, 2001*) being involved in interactions with the pre-mRNA-splicing helicase Brr2 and the ubiquitin-like modifier Hub1 (*van Nues & Beggs, 2001*; *Wilkinson et al., 2004*). Snu66 is a relatively large slightly acidic protein (with pI 6.35) that consists of 587 residues. Figure 5E and Table 1 shows that this protein is predicted to be highly disordered and possesses large number of $\alpha$-MoRFs, clearly indicating that this disordered protein evolved to be involved in a large number of protein–protein interactions. In agreement with this hypothesis, recent study showed that the N-terminal region of Snu66 contains two Hub1 binding motifs, which are highly similar HIND elements (72% identity) arranged in tandem (*Mishra et al., 2011*). The crystal structures of Hub1 in complexes with HIND-I (residues 1-31) and HIND-II (32-62) elements of Snu66 were solved (*Mishra et al., 2011*). Figures 6A and 6B show that both HIND-I and HIND-II elements adopt $\alpha$-helical structure in the bound form, therefore providing experimental support to the $\alpha$-MoRF computationally identified in this region.

*Pre-mRNA-splicing factor Cwc15 (UniProt ID: Q03772).* Cwc15 belongs to the CWC complex (or Cef1-associated complex), which is a spliceosome sub-complex similar to the late-stage spliceosome composed of the U2, U5 and U6 snRNAs and a set of at least 43 spliceosomal proteins, such as Bud13, Brr2, Cdc40, Cef1, Clf1, Cus1, Cwc2, Cwc15,

Cwc21, Cwc22, Cwc23, Cwc24, Cwc25, Cwc27, Ecm2, Hsh155, Ist3, Isy1, Lea1, Msl1, Ntc20, Prp8, Prp9, Prp11, Prp19, Prp21, Prp22, Prp45, Prp46, Slu7, Smb1, Smd1, Smd2, Smd3, Smx2, Smx3, Snt309, Snu114, Spp2, Syf1, Syf2, Rse1, and Yju2. Although the exact function of Cwc15 is still poorly understood, previous studies revealed that this protein positively contributes to Cdc5p/Cef1p function (*Ohi et al., 2002*), suggesting that Cwc15 is potentially associated with the U2 snRNP. Cwc15 is a small highly basic protein (pI 9.06, 175 residues) which is predicted to be highly disordered and contain two $\alpha$-MoRFs, further strengthening its potential role in protein–protein interactions (see Table 1 and Fig. 5F).

*Pre-mRNA-splicing factor Spp381 (UniProt ID: P38282).* Over-expression of Spp381 has been shown to rescue temperature-sensitive mutants of the gene Prp38, which plays an important role is the U4 subunit release from the spliceosome (*Lybarger et al., 1999*). An over-expressed Spp381 however does not rescue a null Prp38 allele, indicating that these two proteins cooperate but are not interchangeable. Spp381 is believed to interact with both the spliceosome and the RNA to be spliced. Immuno-precipitation experiments showed that, similar to Prp38, Spp381 is present in the U4/U6.U5 tri-snRNPs particle and two-hybrid analyses support the view that the C-terminal half of Spp381 directly interacts with the Prp38 protein (*Lybarger et al., 1999*). There is also a putative PEST motif within Spp381, which is one of the hallmarks of IDPs that are known to require tight regulation of their intracellular concentrations (*Singh et al., 2006*). Figure 5G shows that Spp381 (an acidic protein (pI 5.52) consisting of 291 residues) is predicted to be highly disordered and contain 6 potential $\alpha$-MoRFs.

*Pre-mRNA-splicing factor Syf2 (UniProt ID: P53277).* This protein is involved in pre-mRNA splicing and cell cycle control. It is another component of the NTC complex (or Prp19-associated complex), associates to the spliceosome to mediate conformational rearrangement and/or to stabilize the structure of the spliceosome after U4 snRNA dissociation, which leads to spliceosome maturation (*Russell et al., 2000*). Cells with defective Syf2 proteins suffer from cell cycle arrest, possibly due to the inefficient splicing of $\alpha$-tubulin (Tub1) (*Dahan & Kupiec, 2002*). Syf2 was shown to interact with other spliceosomal proteins, such as Cef1, Clf1, Ntc20, Prp19, and Syf1. No crystal structure has been determined as of yet for this protein, and Syf2 is known to possess 4 phosphoserines. Syf2 has 215 residues, pI of 9.34, high level of intrinsic disorder and four $\alpha$-MoRFs (see Table 1 and Fig. 5H).

*Pre-mRNA-splicing factor Cwc26 (UniProt ID: P46947).* This protein belongs to the pre-mRNA retention and splicing complex (*Vincent et al., 2003*), RES, a protein complex that is required for efficient splicing, and prevents leakage of unspliced pre-mRNAs from the nucleus (named for pre-mRNA REtention and Splicing) (*Dziembowski et al., 2004*). In yeast, the complex consists of Ist3p, Bud13p, and Pml1p. It has no posttranslational modification sites and no known crystal structure. It has been shown to interact with the protein Ist3 and Pml1 (*Dziembowski et al., 2004*). Cwc26 is also known as Bud13 protein, since it may also be involved in positioning the proximal bud pole signal (*Zahner, Harkins & Pringle, 1996*; *Ni & Snyder, 2001*; *Vincent et al., 2003*; *Dziembowski et al., 2004*). It has 266

residues and is highly basic (pI 9.31). Its N-terminal half is predicted to be very disordered and is expected to contain two $\alpha$-MoRFs (see Table 1 and Fig. 5I).

*Pre-mRNA-splicing factor Slu7 (UniProt ID: Q02775).* This is an essential protein which is involved in the second catalytic step of the pre-mRNA splicing, participating in the selection of 3'-type splice sites. This selection could be done via a 3'-splice site-binding factor, Prp16 (*Frank & Guthrie, 1992*; *Ansari & Schwer, 1995*; *James, Turner & Schwer, 2002*). The order of recruitment is believed to be Slu7, Prp18 and then Prp22. All three proteins are released from the spliceosome after step 2 concomitantly with the release of mature mRNA. Slu7 protein contains two functionally important domains: a zinc knuckle ($^{122}$CRNCGEAGHKEKDC$^{135}$) and a Prp18-interaction domain ($^{215}$EIELMKLELY$^{224}$) (*Frank & Guthrie, 1992*; *Ansari & Schwer, 1995*; *James, Turner & Schwer, 2002*). It has three phosphoserines and does not have a crystal structure determined. Slu7 consists of 382 residues and is characterized by a pI of 8.89. Figure 5J shows that Slu7 is rather disordered and contains a number of $\alpha$-MoRFs located in its N-terminal half. It is important to emphasize here that two of the predicted $\alpha$-MoRFs (located at regions 111-128 and 213-230) significantly overlap with the aforementioned functional domains of Slu7 protein.

*Protein Cwc16 (UniProt ID : P28320).* Similar to Cwc15 discussed above, Cwc16 (also known as Yju2) is a part of the CWC complex. It was shown that splicing factor Yju2 participates in spliceosome assembly, is associated with the components of the Prp19-associated complex (NineTeen Complex [NTC])) and is required for pre-mRNA splicing (*Liu et al., 2007*). NTC is known to be essential for pre-mRNA splicing, being required for the spliceosome activation by specifying interactions of U5 and U6 with pre-mRNA on the spliceosome after the release of U4. NTC contains at least eight protein components, including two tetratricopeptide repeat (TPR)-containing proteins, Ntc90 and Ntc77 (*Chang, Chen & Cheng, 2009*). Although Yju2 interacts with the spliceosome at almost the same time as NTC during the spliceosome assembly, these two spliceosome components are not entirely in association with each other (*Liu et al., 2007*). Furthermore, Yju2 is not required for the NTC binding to the spliceosome or for NTC-mediated spliceosome activation (*Liu et al., 2007*). However, Yju2 was shown to promote the first catalytic reaction of pre-mRNA splicing after Prp2-mediated structural rearrangement of the spliceosome (*Liu et al., 2007*). It is believed that Yju2 is recruited to spliceosome by the Ntc90 protein (*Chang, Chen & Cheng, 2009*). Cwc16/Yju2 is a medium-size, highly basic protein (pI 9.41, 278 residues) that is predicted to be highly disordered and contain five $\alpha$-MoRFs (see Table 1 and Fig. 5K). Cwc16 is involved in interaction with Syf2 and is predicted to have two nuclear localization signals (NLSs, residues 242-258 and 260-278). Importantly, these NLSs coincide with the two C-terminal $\alpha$-MoRFs.

*Pre-mRNA-splicing factor Ntr2 (UniProt ID: P36118).* Ntr2 is a part of the NTR complex (NTC-related complex), which is composed of Ntr1, Ntr2 and Prp43. Ntr2 is known to interact with Clf1, Ntr1 and Prp43, and, along with Ntr1, is involved in the pre-mRNA splicing and spliceosome disassembly, promoting the release of excised intron from the spliceosome by acting as a receptor for Prp43, possibly assisted by the Ntr1 protein

(*Tsai et al., 2005*; *Boon et al., 2006*). This specific Prp43 targeting leads to the disassembly of the spliceosome with the separation of the U2, U5, U6 snRNPs and the NTC complex (*Tsai et al., 2005*; *Boon et al., 2006*). Ntr2 has two phosphoserines and no known crystal structure. This is a medium-size acidic protein (pI 5.51, 322 residues) that is predicted to be very disordered and to contain three $\alpha$-MoRFs (see Table 1 and Fig. 5L).

*Nucleolar protein 3 (UniProtID: Q01560).* Npl3 contains two RRM (RNA recognition motifs) at the positions 125-195 and 200-275, indicating that it interacts directly with the Poly(A) regions mRNA (*Wilson et al., 1994*; *Burkard & Butler, 2000*). It has 5 phosphoserines and Arg/Gly-rich region at position 280-398. Nlp3 can interact with the riboexonuclease Rrp6, which plays a role in 5.8S rRNA 39-end processing and whose defective mutants suppress the growth defect associated with an mRNA polyadenylation defect (*Burkard & Butler, 2000*). Npl3 consists of 414 residues and has a pI of 5.38. It is predicted to be mostly disordered and is expected to contain five $\alpha$-MoRFs (see Table 1 and Fig. 5M). Solution structures of two domains containing RRMs (residues 114-201 and 193-282) have been determined using a novel expressed protein ligation protocol (*Skrisovska & Allain, 2008*). The resulting structures are shown in Figs. 6C and 5D.

*Pre-mRNA-splicing factor Spp2 (UniProt ID: Q02521).* Pre-mRNA processing occurs by assembly of splicing factors on the substrate pre-mRNA to form the spliceosome followed by two consecutive RNA cleavage-ligation reactions. The Spp2 protein belongs to the CWC complex (or CEF1-associated complex) and interacts with Prp2 (*Silverman et al., 2004*). Spp2 is important for the pre-mRNA splicing, playing a role at the final stages of the spliceosome maturation by promoting the first step of splicing (*Roy et al., 1995*). Although this first reaction is controlled by the Prp2 protein that hydrolyzes ATP, a model was proposed in which Spp2 binds to the spliceosome complex I (composed of mRNA, U1, U2, U4, U5, and U6 smRNPs) in the absence of Prp2p or ATP. This would be followed by Prp2p binding and subsequent ATP hydrolysis leading to the catalytic reaction resulting in the formation of complex II and the release of both proteins from the spliceosome (*Roy et al., 1995*). The Spp2 protein has one phosphoserine and no known crystal structure. Spp2 is a small moderately basic protein (pI 8.79, 185 residues) that possesses a G-patch domain (residues 100-149) and is predicted to have one $\alpha$-MoRF and be mostly disordered (see Table 1 and Fig. 5N).

*Bud site selection protein 31, Bud31 (UniProt ID: P25337).* Bud31 is one of the NTC-related proteins which also a component of the Cef1p sub-complex. Although it is better known for its role in the bud site selection in yeast replication, Bud31 also appears to play a role in the yeast spliceosome through interaction with the protein Cef1, as well as interaction with the precatalytic B complex, and interaction with catalytically active complexes with stably bound U2, U5, and U6 smRNPs (*Saha et al., 2012b*). Recently, Bud31 was shown to be important for the efficient progression to the first catalytic step and to be required for the second catalytic step in reactions at higher temperatures (*Saha et al., 2012b*). Bud31 plays a role in both cell cycle transitions and pre-mRNA splicing. It was shown recently that Bud31 promotes transition through the G1-S regulatory point (Start) but is not needed for G2-M transition or for exit from mitosis (*Saha et al., 2012a*). By analyzing the splicing status of

transcripts that encode proteins involved in yeast budding, Bud31 was shown to facilitate the efficient splicing of only some of these pre-mRNAs (*Saha et al., 2012a*). Bud31 is a small basic protein (pI 9.64, 157 residues) that contains an N-terminally located NLS (residues 2-11), has no posttranslational modification sites and no known crystal structure. This protein is predicted to be moderately disordered and to possess one $\alpha$-MoRF (see Table 1 and Fig. 5O).

*smRNP-associated protein B, SmB (UniProt ID: P40018).* SmB protein is also referred to as snRNP-associated protein B, snRNP-B. SmB is involved in pre-mRNA splicing, along with other Sm core proteins: SmB', SmD1, SmD2, SmD3, SmE, SmF, and SmG. It binds to U1, U2, U4, U5 snRNA, all containing a highly conserved region, referred to as the Sm binding site. It belongs to the SmB and SmN family, and is located in the cell nucleus. Sm core proteins have an important role during the formation of snRNPs. The SmB protein is an important part of the Sm core complex, as it is found in immunoprecipitates of U1, U2, U4, and U5 snRNAs (*Camasses et al., 1998*). Along with other Sm proteins, SmB contains a common sequence motif, which helps forming the globular core of the spliceosome snRNPs (U1, U2, U5, and U4/U6) (*Walke et al., 2001*). SmB possesses a nuclear localization signal (NLS) located in the C-terminal half of the protein (region 105-132). When this portion of the sequence is either deleted or mutated, SmB function is lost, suggesting that the C-terminal part of this Sm protein has been evolutionary conserved, and its function determines nuclear localization (*Bordonne, 2000*). This protein consists of 196 residues, has a pI of 10.37, contains one $\alpha$-MoRF, and shows high levels of disorder, especially in it C-terminal part (see Table 1 and Fig. 5P). When analyzed by seven disorder predictors, PONDR® FIT, PONDR® VLXT, PONDR® VL3, PONDR® VSL2B, IUPred, Foldindex, and TopIDP, its corresponding levels of disorder are 0.643, 0.648, 0.724, 0.760, 0.571, 0.628, and 0.719, respectively.

*U1 snRNP protein C, Yhc1 (UniProt ID: Q05900).* Yhc1 (also known as U1-C protein) is an important component of the spliceosome subcomplex U1 snRNP (*Tang et al., 1997*), which is composed of the 7 core Sm proteins common to all spliceosomal snRNPs, and at least 10 particle-specific proteins (see Table 1 and Fig. 4), and which is essential for recognition of the pre-mRNA 5' splice-site and the subsequent assembly of the spliceosome (*Fabrizio et al., 2009*). The major functional role of Yhc1 is the initial 5' splice-site recognition for both constitutive and alternative splicing. Yhc1 interacts with the U1 snRNA and the 5' splice-site region of the pre-mRNA, therefore stimulating the commitment complex formation by stabilizing the base pairing of the 5' end of the U1 snRNA and the 5' splice-site region (*Tang et al., 1997*; *Zhang & Rosbash, 1999*). It was shown that Yhc1 can recognize the 5' splice-site in the absence of base-pairing between the pre-mRNA and the U1 snRNA (*Du & Rosbash, 2002*). Yhc1 is a highly basic protein (pI 10.11) that consists of 231 residues and contains a matrin-type zinc finger domain (residues 4-36). Yhc1 is predicted to be moderately disordered and is expected to contain two $\alpha$-MoRFs (see Table 1 and Fig. 5Q).

*U2 snRNP protein Cus1 (UniProt ID: Q02554).* Cus1, also known as cold sensitive U2 snRNA suppressor, is a 436 residues long protein that is required for the U2 snRNP binding

to pre-mRNA during spliceosome assembly (*Pauling, McPheeters & Ares, 2000*). Cus1 is a homologue of the human Sap145 protein that is present in the 17S form of the human U2 snRNP. Yeast Cus1 interacts with U2 snRNA, with Hsh49 via the 82-amino-acid-long region located between positions 229 and 311 and with Hsh155 (*Pauling, McPheeters & Ares, 2000*). Based on these observations it was proposed that Cus1, Hsh49, and Hsh155 form a stable protein complex which can exchange with a core U2 snRNP and which is necessary for U2 snRNP function in pre-spliceosome assembly (*Pauling, McPheeters & Ares, 2000*). Although Cus1 is a moderately basic protein (pI 8.67), one of its characteristic features is a highly acidic nature of its C-terminal tail, where nearly half of the last 59 residues are acidic (23 are E or D) (*Pauling, McPheeters & Ares, 2000*). Both N-terminal and C-terminal tails of Cus1 are predicted to be highly disordered and contain a number of potential disorder-based binding sites (see Table 1 and Fig. 5R).

*U5 snRNP protein Lin1 (UniProt ID: P38852).* Lin1 is a multifunctional protein involved in several different processes. Compartmentalization of Lin1 with U5 snRNP was inferred from a direct assay (*Stevens et al., 2001*). Based on its association with the Irr1/Scc3 component of the cohesin complex involved in cohesion and separation of chromosomes during mitosis and its interaction with Prp8, Slx5, Siz2, Wss1, Rfc1, and YIL149w proteins, which are known to participate in mRNA splicing, DNA replication, chromosome condensation, chromatid separation and alternative cohesion, Lin1 was proposed to serve as a functional and physical link among these processes (*Bialkowska & Kurlandzka, 2002*). Lin1 is an acidic protein (pI 5.01) consisting of 340 residues. Figure 5S show that the N-terminal half of the Lin1 protein is predicted to be very disordered and is expected to have four $\alpha$-MoRFs (see also Table 1), whereas the C-terminal half is expected to be ordered. The last sixty residues of Lin1 (residues 282-340) correspond to a glycine-tyrosine-phenylalanine (GYF) domain which contains a conserved GP[YF]xxxx[MV]xxWxxx[GN]YF motif which can be involved in the recognition of proline-rich sequences (*Freund et al., 1999*). Since many proline-rich proteins are IDPs, Lin1 utilizes two different modes of intrinsic disorder-based protein–protein recognition, where it relies on the intrinsic disorder of its N-terminal half to interact with some partners and also uses intrinsic disorder of other partners to interact with ordered C-terminal region.

*U4/U6 snRNP protein Prp3 (UniProt ID: Q03338).* Prp3 is large moderately basic protein (pI 8.69, 469 residues), which is a component of the yeast U4/U6 snRNP and is also present in the U4/U6.U5 tri-snRNP (*Anthony, Weidenhammer & Woolford, 1997*). It was shown that Prp3 is necessary for both the formation of stable U4/U6 snRNPs and for the assembly of the U4/U6.U5 tri-snRNP from its component snRNPs. In fact, the Prp3 inactivation diminishes the spliceosome assembly from the pre-spliceosome due to the absence of intact U4/U6.U5 tri-snRNPs (*Anthony, Weidenhammer & Woolford, 1997*). Homology between the yeast Prp3 protein and the human protein 90K (which is a component of the human U4/U6 snRNPs) represents an illustrative example of the conservation of splicing factors between yeast and metazoans (*Anthony, Weidenhammer & Woolford, 1997*). Prp3 is predicted to contain significant amount of disorder (especially in its first 350 residues) and is expected to be a promiscuous binder, since it has seven $\alpha$-MoRFs (see Table 1 and Fig. 5T).

*U6 snRNA-associated Sm-like Protein LSm4 (UniProt ID: P40070).* Sm-like (LSm) heptameric complex is one of the important spliceosomal components, which exists in two different forms, the nuclear form and the cytoplasmic form, each comprising of different subunits (*Reijns, Auchynnikava & Beggs, 2009*). The nuclear form, LSm2-8 complex, consists of subunits from LSm2 to LSm8, is closely associated with the U6 snRNP, interacts with the Prp24, and works together with the neighboring proteins to create a functional spliceosome. The cytoplasmic form is the composed of LSm1 to LSm7 and is involved in mRNA turnover and also promotes the mRNA decapping and decay (*Spiller et al., 2007*). One of the roles of the LSm2-8 complex is to promote the U4/U6 di-snRNP assembly (*Reijns, Auchynnikava & Beggs, 2009*). It is also involved in the processing and stabilization of ribosomal RNAs and determines the nuclear localization of the U6 snRNP (*Spiller et al., 2007*). LSm4 is a component of both LSm1-7 and LSm2-8 complexes. Among different functions ascribed to LSm4 are specific binding to the 3'-terminal U-tract of U6 snRNA, participation in processing of pre-tRNAs, pre-rRNAs and U3 snoRNA, and involvement in maturing of the precursor of the RNA component of RNase P (pre-P RNA) (*Bouveret et al., 2000*; *Tharun et al., 2000*; *Kufel et al., 2002*; *Kufel et al., 2003*; *Kufel et al., 2004*). LSm4 is a small basic protein (pI 9.45, 187 residues) with highly disordered C-terminal domain that contains one $\alpha$-MoRF and one phosphoserine at position 181 (*Albuquerque et al., 2008*) (see Table 1 and Fig. 5U).

*Early splicing factor Prp5 (UniProt ID: P21372).* Prp5 is a large slightly basic (pI 8.22) ATP-dependent RNA helicase consisting of 850 residues (*O'Day, Dalbadie-McFarland & Abelson, 1996*). Prp5 is involved in spliceosome assembly, nuclear splicing, and catalysis of the ATP-dependent conformational change of U2 snRNP (*Ruby, Chang & Abelson, 1993*; *Wells & Ares, 1994*; *O'Day, Dalbadie-McFarland & Abelson, 1996*; *Abu Dayyeh et al., 2002*). It is believed that this protein might be involved in bridging U1 and U2 snRNPs and might promote stable interaction between the U2 snRNP and intron RNA (*Xu et al., 2004*). Prp5 contains a helicase domain (residues 287-661) which is divided in the helicase ATP-binding and helicase C-terminal subdomains (residues 287-467 and 502-661, respectively). There are also several functionally important motifs in Prp5, such as nucleotide binding motif (residues 300-307), coiled-coil (residues 13-81), NLS (residues 90-96), Q motif (residues 255-284) and the DEAD-box motif (residues 415-418). Despite the fact that Prp5 is an enzyme and therefore is expected to be mostly ordered, Table 1 and Fig. 5V shows that this protein is predicted to have significant amount of disorder (mostly located in the first N-terminal 200 residues) and also to possess six $\alpha$-MoRFs.

*CBP protein Cbc2 (UniProt ID: Q08920).* Cbc2 is a component of the nuclear cap-binding complex (CBC), which is a heterodimer that co-transcriptionally interacts with the cap of pre-mRNAs and is composed of the Sto1/Cbc1 and Cbc2 proteins. CBC complex is crucial for the efficient pre-mRNA splicing through its participation in the formation of the commitment complex and spliceosome. It is involved in maturation, export and degradation of nuclear mRNAs (*Lewis, Gorlich & Mattaj, 1996*; *Fortes et al., 1999*). Cbc2 binds the m7G cap of the RNA and a large CBC subunit Sto1 that interacts with

karyopherins, and is believed to be responsible for splicing control during meiosis (*Qiu et al., 2012*). Cbc2 is an acidic protein (pI 5.02) that is composed of 208 residues and contains RRM domain that is involved in single-stranded RNA binding (residues 46-124) and three mRNA cap-binding regions (residues 118-122, 129-133, and 139-140). Figure 5W shows that Cbc2 is predicted to have long disordered tails and two $\alpha$-MoRF located within these intrinsically disordered N- and C-termini (see also Table 1).

*Msl5 protein (UniProt ID: Q12186)*. Msl5 is the branch point-bridging protein, which is required for the pre-spliceosome formation, playing a role in the creation of the commitment complex 2 (CC2) where it binds to the snRNP U1-associated protein Prp40, bridging the U1 snRNP-associated 5'-splice site and the Msl5-associated branch point 3' intron splice site (*Abovich & Rosbash, 1997*; *Rutz & Seraphin, 1999*). As a part of the CC2 complex, Msl5 is involved in the nuclear retention of pre-mRNA (*Rutz & Seraphin, 2000*). It interacts with Mud2 and Prp40 (*Abovich & Rosbash, 1997*; *Rutz & Seraphin, 1999*), and the proline-rich region of Msl5 (residues 363-474) binds to the GYF domains of Smy2 and Syh1 (*Kofler, Motzny & Freund, 2005*). Figure 5X shows that the Msl5 region responsible for the interaction with the GYF domains of Smy2 and Syh1 is a part of the long, highly disordered tail. There are two $\alpha$-MoRFs in this basic (pI 9.72), 476 residue-long protein (see Table 1 and Fig. 5X).

## Highly disordered spliceosomal proteins might act as important hubs

Protein-protein interaction networks contain many proteins with only a few links and a few proteins with many links. These highly connected or promiscuous proteins are known as hubs, the binding mechanisms of which can be reasonably explained based on the molecular recognition via disorder-to-order transitions upon binding (*Dunker et al., 2005*). With respect to timing issues, some proteins have multiple, simultaneous interactions ("party hubs") (*Han et al., 2004*) while others have multiple sequential interactions ("date hubs") (*Han et al., 2004*). Perhaps date hubs connect biological modules to each other (*Hartwell et al., 1999*) while party hubs form scaffolds that enable the assembly of functional modules (*Silverman et al., 2004*). The overall importance of intrinsic disorder for function of hub proteins was analyzed in several recent bioinformatics publications (*Dosztanyi et al., 2006*; *Ekman et al., 2006*; *Haynes et al., 2006*; *Patil & Nakamura, 2006*; *Singh et al., 2006*). Disorder appears to be more clearly associated with date hubs (*Ekman et al., 2006*; *Singh et al., 2006*) than with party hubs. However, some protein complexes clearly use long regions of disorder as a scaffold for assembling an interacting group of proteins (*Hohenstein & Giles, 2003*; *Jaffe, Aspenstrom & Hall, 2004*; *Luo & Lin, 2004*; *Rui et al., 2004*; *Wong & Scott, 2004*; *Jaffe & Hall, 2005*; *Marinissen & Gutkind, 2005*; *Salahshor & Woodgett, 2005*; *Carpousis, 2007*).

Due to their malleable nature, IDPs and IDPRs are predisposed to be hubs. In fact, they are commonly involved in one-to-many and in many-to-one binding scenarios. Both of these interaction modes are specific cases of the date hubs, which can bind different proteins, but not at the same time. In the first mechanism, one unfolded segment is used by

a protein to interact with multiple unrelated binding partners. In the second mechanism, many unrelated unfolded fragments are used by unrelated proteins to interact with the same partner (*Dunker et al., 1998*; *Oldfield et al., 2008*).

   To check the set of highly disordered spliceosomal proteins for "hubness", we utilized the STRING database, which acts as a 'one-stop shop' for all information on functional links between proteins (*Szklarczyk et al., 2011*). Version 9.0 of STRING (accessible at http://string-db.org) covers more than 1100 completely sequenced organisms, including *Saccharomyces cerevisiae*. Figure 7 represents results of the STRING'ing for the 24 yeast spliceosomal proteins considered in a previous section. Here, the interactome of each of these proteins is shown as an interaction network, where proteins are represented by spheres (note that in each network, the red sphere corresponds to a query protein) and connections between two proteins are shown by lines. The fundamental unit stored in STRING is the "functional association"; i.e., the specific and biologically meaningful functional connection between two proteins (*Szklarczyk et al., 2011*). These functional associations are based on the seven types of evidence, such as fusion evidence, neighborhood evidence, co-occurrence evidence, experimental evidence, text mining evidence, database evidence, and co-expression evidence (*Szklarczyk et al., 2011*). These different types of evidence are shown by the lines of different color. It is necessary to emphasize that Fig. 7 is used here with a strictly illustrative purpose; i.e., to show that all of the analyzed spliceosomal proteins are involved in multiple interactions and therefore can be considered as hubs. Since these 24 proteins contain significant amount of predicted disorder and since almost all of them interacts with other spliceosomal proteins many of which are also predicted to be mostly disordered, Fig. 7 suggests that hubness of spliceosomal proteins is related to their intrinsically disordered nature and/or by the intrinsic disorder of their partners.

## CONCLUDING REMARKS

In this work we have studied the prevalence of intrinsic disorder in the yeast spliceosome in order to test if this complex ribonucleoprotein machine had an enhanced predisposition for intrinsic disorder in comparison with the average proteome. Our results showed that the prevalence of IDPs/IDPRs in the spliceosome was not significantly different from the averaged disorderedness of the eukaryotic proteins. However, being compared with the behavior of an averaged yeast protein, yeast spliceosomal proteins were noticeably more disordered. For example, 46.7% of the spliceosomal proteins were shown to be mostly disordered, whereas the entire yeast proteome contained significantly smaller amount of such proteins (13.3%). Furthermore, ~61% spliceosomal proteins were shown to possess $\alpha$-MoRFs, while there were 21.1% of MoRF-containing proteins in the entire yeast proteome. This suggests that the spliceosomal proteins are often engaged in interactions with their protein and RNA partners via disordered regions. More detailed analysis of the most disordered spliceosomal proteins revealed that they are in fact involved in multiple interactions and therefore can be considered as disordered hubs.

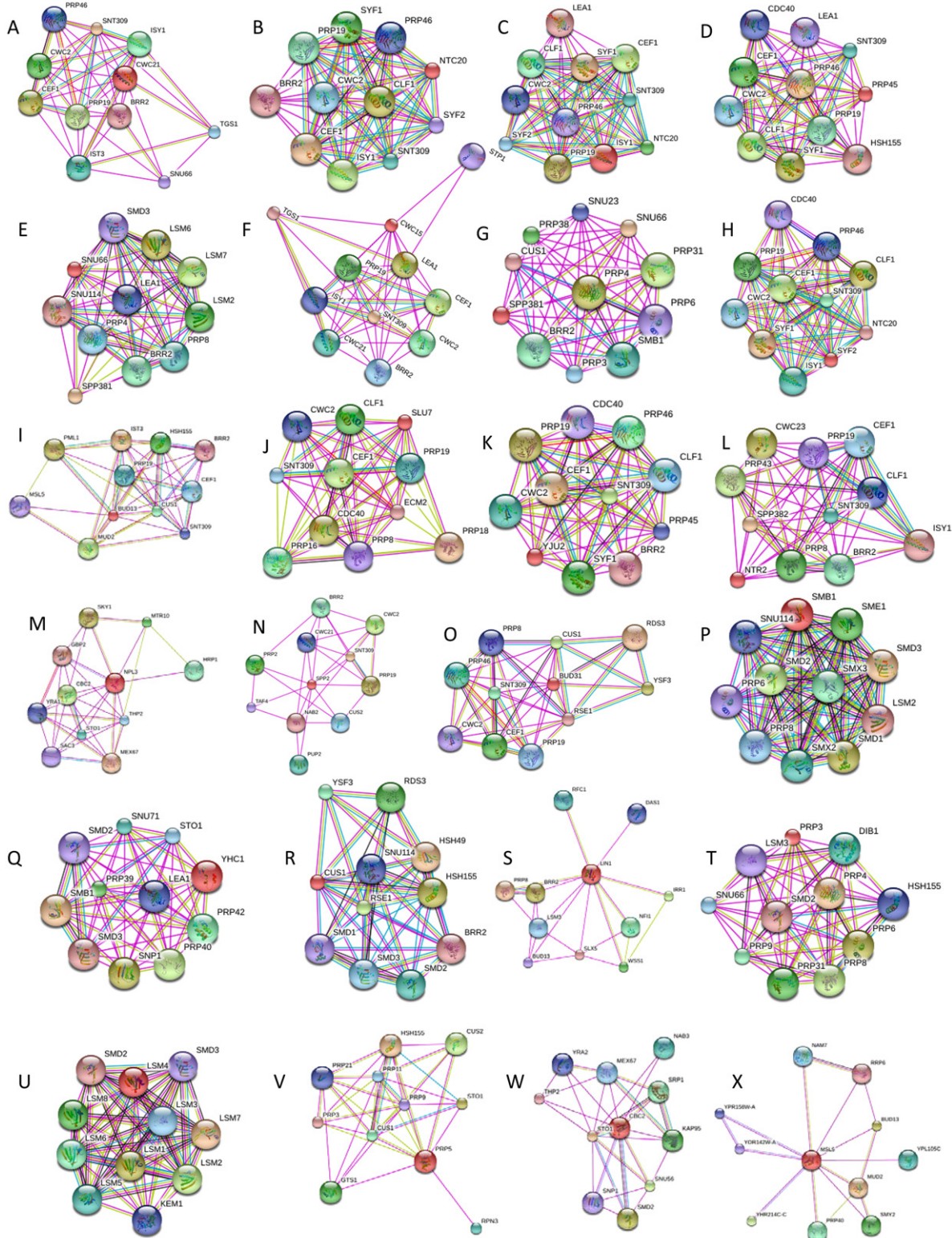

**Figure 7 STRING analysis of the interactomes of illustrative spliceosomal proteins.** A. Cwc21; B. Ntc20; C. Isy1/Ntc30; D. Prp45; E. Snu66; F. Cwc15; G. Spp381; H. Syf2; I. Cwc26; J. Slu7; K. Yju2/Cwc16; L. Ntr2; M. Npl3; N. Spp2; O. Bud31; P. SmB; 

**Figure 7 (...continued)**
Q. Yhc1; R. Cus1; S. Lin1; T. Prp3; U. LSm4; V. Prp5; W. Cbc2; and X. Msl5. STRING database is the online database resource Search Tool for the Retrieval of Interacting Genes, which provides both experimental and predicted interaction information (*Szklarczyk et al., 2011*). For each protein, STRING produces the network of predicted associations for a particular group of proteins. The network nodes are proteins. The edges represent the predicted functional associations. An edge may be drawn with up to 7 differently colored lines – these lines represent the existence of the seven types of evidence used in predicting the associations. A red line indicates the presence of fusion evidence; a green line – neighborhood evidence; a blue line – co-occurrence evidence; a purple line – experimental evidence; a yellow line – text mining evidence; a light blue line – database evidence; a black line – co-expression evidence (*Szklarczyk et al., 2011*).

Our findings are in a good agreement with the earlier published results on the peculiarities of intrinsic disorder distribution and functions in known human spliceosomal proteins (*Korneta & Bujnicki, 2012*). The authors of that study concluded that about half of the residues in the human spliceosomal proteome are expected to be intrinsically disordered. Furthermore, a correlation was found between the type of protein disorder and its function and localization within the spliceosome, with the spliceosomal components involved in earlier stages of the splicing process being more disordered than components acting at the later stages (*Korneta & Bujnicki, 2012*). This enrichment of early proteins in disorder was proposed to play a significant functional role, since proteins of the components of the spliceosome that act earlier in the process are crucial for the establishing a network of interactions (*Korneta & Bujnicki, 2012*). In agreement with these conclusions Fig. 4 and Table 1 show that yeast spliceosomal proteins related to the complex B are expected to be more disordered than proteins related to the spliceosomal components engaged at the later stages. Therefore, intrinsic disorder is abundant in the yeast spliceosome and is important to assembly and action of this malleable ribonucleoprotein machine.

### Funding

This work was supported in part by the Programs of the Russian Academy of Sciences for the "Molecular and Cellular Biology" (to VNU). The funders had no role in study design, data collection and analysis, decision to publish, or preparation of the manuscript.

### Grant Disclosures

The following grant information was disclosed by the authors:
Programs of the Russian Academy of Sciences for the "Molecular and Cellular Biology".

### Competing Interests

Vladimir N. Uversky and Bin Xue are Academic Editors for PeerJ. We do not have other competing interests.

### Author Contributions

- Maria de Lourdes Coelho Ribeiro and Julio Espinosa performed the experiments, analyzed the data, wrote the paper.

- Sameen Islam, Osvaldo Martinez, Jayesh Jamnadas Thanki, Stephanie Mazariegos and Tam Nguyen performed the experiments, wrote the paper.
- Maya Larina performed the experiments.
- Bin Xue performed the experiments, analyzed the data, contributed reagents/materials/analysis tools.
- Vladimir N. Uversky conceived and designed the experiments, performed the experiments, analyzed the data, contributed reagents/materials/analysis tools, wrote the paper.

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
