# Peer review of "Malleable ribonucleoprotein machine: protein intrinsic disorder in the Saccharomyces cerevisiae spliceosome"

_PeerJ, doi:10.7717/peerj.2_

## Round 0.1 · original submission · Minor Revisions

· Academic Editor

Minor Revisions

Please consider carefully the comments and criticisms of the reviewers as they can help to improve considerably your manuscript.

·

Basic reporting

This artcile meets the required standard for publication in PeerJ

Experimental design

The experimental design is robust. The use of several complementary approaches adds to the robustness of the study.

Validity of the findings

Obtained results were properly interpreted and discussed. Results convincingly argue for the major conclusion drawn by the authors (ie abundance of disorder in S. cerevisiae splicesome)

Additional comments

Report for MS entitled "Malleable Ribonucleoprotein Machine: Protein Intrinsic Disorder in the Saccharomyces cerevisiae Spliceosome" submitted to PeerJ.

In this paper, Ribeiro et al have carried out an in-depth computational analysis of the S. cerevisiae splicesome. By combining various predictors of disorder, relying on different philosophies, they obtained clear evidence for the abundance of disorder in this complex ribonucleoprotein machine. In addition, they carried out an extensive functional (ie bibliographic) description of a selection of the most disordered proteins.
Altogether this work is technically sound and impressive in terms of the quantity and quality of complementary in silico analyses that were carried out. Given the growing interest that IDPs are being paid, this work represents an important contribution to the field that is expected to have a noticeable impact. As such, it merits to be published in PeerJ. A few points (listed below) need however to be addressed before the MS can be accepted for publication.


Minor points
- A typo in the title and in the legend of Table 1 should be corrected (eg cerivisiae that should be cerevisiae)
- Line 699. I do not agree with the conclusion of the authors that Npl3 is predicted to be disordered based on Figure 5M. At best it could be said that is predicted to be mostly disordered. Indeed the central part is not predicted to be disordered (see PONDR scores well below 0.5) and indeed this part encompasses the two domains the structure of which was solved by NMR.
- Lines 702-705. Although I agree with the authors that protein ligation is an interesting and promising strategy, I do not think that the description of this method merits to be included in this MS.
- Line 423. Beyond the paper by He et al, the papers Ferron et al Proteins 2006 and Bourhis et al Curr Prot and Pept Sci 2007 merit to be cited when the authors refer to the advantage of combining various disorder predictors relying on different principles.
- Point about the agreement (eg linear relationship) between PONDR-FIT and PONDR-VLXT. A linear relationship was to be expected since PONDR-FIT is a metaserver that includes PONDR VLXT. The results shown in Figure 3B are much more suited to make the desired point as they were obtained using truly independent approaches. The authors may want to add a comment on that.
-Figure 4. What does the grey stand for?

Cosmetic points

-Abstract and line 203. The expression « prevalence of intrinsic disorder » would be preferable to the expression « prevalence of intrinsically disordered proteins » as the authors show the abundance of disorder (ie disordered regions) in the splicesome rather than the abundance of fully disordered proteins.
-Following introduction of the IDP acronym, this latter should be used rather than the extended term of “intrinsically disordered proteins” (see lines 147-149 for instance).
-Line 194. « of » should be added between « abundance » and « intrinsic disorder »
-Line 229. Suppress « or » after « set of » . Add « of » before « disordered … ». Replace « form » with « from ».
-Line 234. Suppress « rigid » . Replace « DNA » by « nucleic acids ».
-Line 313 and elsewhere. The use of the expression « binding to partners » is preferable to « binding to partner sequences » as this latter may wrongly suggests that the IDP necessarily recognizes a linear motif in the partner.
-Line 316. Replace « 24 the most » by « the 24 most ».
-Line 371. Replace « ..with that … » by « ..with the prevalence observed for … »
-Line 395. I guess PONDR–FIT is meant.
-Line 413. Suppress « the majority ».
-Line 432 and elsewhere. If it is clear that proteins have been classified based on their average disorder score (<10%, > 10% and > 30%), it is not clear what “30%>IDP” means.
-Line 459. Suppress “the” before “mass” and “in”.
Lines 480-483. This part shoud be suppressed as already said in Materials and Methods.
-Line 546. Replace “phenotype” by “default” for the sake of clarity.
-Line 585. Replace “interact” by “interactions”.
-Line 590. Correct the typo (ie replace “resent” by “recent”)
-Line 605. Replace “the analysis” by “previous studies” as the point that is being made here results from studies that were not carried out by the authors of the present work.
-Line 651. Replace “recruiting” by “recruitment”.
-Line 652. Replace “concomitant” by “concomitantly”.
-Line 656. Suppress “is” before “consists”.
-Line 683. Replace “interacts” with “interact”.
-Line 717. The term “fit” should be suppressed I guess.
-Line 741. Replace “is part of” by “is involved in”
-Line 743. It is not clear to what “which” does refer to. Sm core? SmB?
-Line 755. Replace “espevially” by “especially”.
-Line 851. Replace “concentrated in … residues” by “located in the first 200 N-terminal residues”
-Line 922. Replace “expected” by “predicted” and add “mostly” before “disordered”.
-Line 923. Replace “defined by” with “related to”.
-Lines 944-952. Replace “parts” by “components”.

·

Basic reporting

The paper titled "Malleable Ribonucleoprotein Machine: Protein Intrinsic Disorder in the Saccharomyces cerivisiae Spliceosome" describing the presence, frequency and possible roles of protein disorder in yeast spliceosome. While the paper is well written in general, and have the scientific significacne to publish in PeerJ, several point need to be improved before final acceptance.
- As a basic problem, i would reccomend to shorten the introduction at least to the half of the current size, it contains many interesting, but unrelated informaton to the scientific problem discussed in the paper.

Experimental design

- The experimental design is almost perfect in the paper, and is absolutely acceptable in its present form, but i reccomend to improve the molecular recognition section with the involvment of a different predictor. As it is written in the disorder prediction section, using different predictors (with different calculation methods) resulted in more reliable results. I think, that using ANCHOR predictor would further improve the recognition motif results.

Validity of the findings

- The biggest problem with the manuscript is the scientifically insufficient interpretation of the results. The trems "moderately depleted", "moderately enriched", "rather common", "clearly underrepresented", "being also a bit higher", etc. have no meaning in a scientific paper. All differences found in the experience must be analyzed by the appropriate statistical methods. These statistical analyses are required in the residue composition, long disordered region and MORF content comparisons.

Additional comments

- The much lower number of proteins in yeast spliceosome compared to the human suggests a different, or maybe a simplified splicing mechanism in yeast. Is it possible, that this reduction is related to the extremely low number of spliceable genetic material (only about 250 introns can be found in S. cerevisiae)? While the splicing is required to the survival of the yeast, could it be possible, that it is evolutionary simplified genetically and also in its proccessing?
- If the sequential homology is rather low between yeast and human spliceosomal proteins, can you find any higher level homology (e.g. conserved length, charge bias, conserved disorder pattern)?

---

## Round 0.2 · accepted · Accept

· Academic Editor

Accept

Thank you for carefully replying to the reviewers and improving the manuscript. Good work!